



# Simulation-based comparison of multivariate ensemble post-processing methods

Sebastian Lerch[1], Sándor Baran[2], Annette Möller[3], Jürgen Groß[4], Roman Schefzik[5], Stephan Hemri[6], and Maximiliane Graeter[1]

[1]Karlsruhe Institute of Technology, Karlsruhe, Germany
[2]University of Debrecen, Debrecen, Hungary
[3]Technical University of Clausthal, Clausthal, Germany
[4]University of Hildesheim, Hildesheim, Germany
[5]German Cancer Research Center (DKFZ), Heidelberg, Germany
[6]Federal Office of Meteorology and Climatology MeteoSwiss, Zurich-Airport, Switzerland

**Correspondence:** Sebastian Lerch (Sebastian.Lerch@kit.edu)

**Abstract.** Many practical applications of statistical post-processing methods for ensemble weather forecasts require to accurately model spatial, temporal and inter-variable dependencies. Over the past years, a variety of approaches has been proposed to address this need. We provide a comprehensive review and comparison of state of the art methods for multivariate ensemble

post-processing. We focus on generally applicable two-step approaches where ensemble predictions are first post-processed separately in each margin, and multivariate dependencies are restored via copula functions in a second step. The comparisons are based on simulation studies tailored to mimic challenges occurring in practical applications and allow to readily interpret the effects of different types of misspecifications in the mean, variance and covariance structure of the ensemble forecasts on the performance of the post-processing methods. Overall, we find that the Schaake shuffle provides a compelling benchmark

that is difficult to outperform, whereas the forecast quality of parametric copula approaches and variants of ensemble copula coupling strongly depend on the misspecifications at hand.

## 1   Introduction

Despite continued improvements ensemble weather forecasts often exhibit systematic errors that require correction via statis-

tical post-processing methods. Such calibration approaches have been developed for a wealth of weather variables and specific applications. The employed statistical techniques include parametric distributional regression models (Gneiting et al., 2005; Raftery et al., 2005) as well as nonparametric approaches (Taillardat et al., 2016) and semi-parametric methods based on modern machine learning techniques (Rasp and Lerch, 2018). We refer to Vannitsem et al. (2018) for a general overview and review.





While much of the developments have been focused on univariate methods, many practical applications require to accurately capture spatial, temporal or inter-variable dependencies (Schefzik et al., 2013). Important examples include hydrological applications (Scheuerer et al., 2017), air traffic management (Chaloulos and Lygeros, 2007) and energy forecasting (Pinson and Messner, 2018). Such dependencies are present in the raw ensemble predictions, but are lost if standard univariate post-processing methods are applied separately in each margin.

Over the past years, a variety of multivariate post-processing methods has been proposed, see Schefzik and Möller (2018) for a recent overview. Those can roughly be categorized into two groups of approaches. The first strategy aims to directly model the joint distribution by fitting a specific multivariate probability distribution. This approach is mostly used in low-dimensional settings, or if a specific structure can be chosen for the application at hand. Examples include multivariate models for temperatures across space (Feldmann et al., 2015), for wind vectors (Schuhen et al., 2012; Lang et al., 2019b), and joint

models for temperature and wind speed (Baran and Möller, 2015, 2017).

The second group of approaches proceeds in a two-step strategy. In a first step, univariate post-processing methods are applied independently in all dimensions, and samples are generated from the obtained probability distributions. In a second step, the multivariate dependencies are restored by re-arranging the univariate sample values with respect to the rank order structure of a specific multivariate dependence template. Mathematically, this corresponds to the application of a (parametric

or non-parametric) copula. Examples include ensemble copula coupling (Schefzik et al., 2013), the Schaake shuffle (Clark et al., 2004) and the Gaussian copula approach (Möller et al., 2013).[1]

Here, we focus on this second strategy which is more generally applicable in cases where no specific assumptions on the parametric structure can be made, or where the dimensionality of the forecasting problem is too high to be handled by fully parametric methods. The overarching goal of this paper is to provide a systematic comparison of state of the art methods

for multivariate ensemble post-processing. In particular, our comparative evaluation includes recently proposed extensions of the popular ensemble copula coupling approach (Hu et al., 2016; Ben Bouallègue et al., 2016). We propose four simulation settings which are tailored to mimic different situations and challenges that arise in applications of post-processing methods. In contrast to case studies based on real-world datasets, simulation studies allow to specifically tailor the multivariate properties of the ensemble forecasts and observations, and to readily interpret the effects of different types of misspecifications on the

forecast performance of the various post-processing methods. Simulation studies have been frequently applied to analyze model properties, and to compare modeling approaches and verification tools in the context of statistical post-processing, see, e.g., Williams et al. (2014); Thorarinsdottir et al. (2016); Wilks (2017); Allen et al. (2019).

The remainder is organized as follows. Univariate and multivariate post-processing methods are introduced in Section 2. Section 3 provides descriptions of the four simulation settings, with results discussed in Section 4. The paper closes with a

discussion in Section 5. Technical details on specific probability distributions and multivariate evaluation methods are deferred

---

[1]An alternative post-processing approach that allows to preserve multivariate dependencies is the member-by-member method proposed by Van Schaeybroeck and Vannitsem (2015). Schefzik (2017) demonstrates that member-by-member post-processing can be interpreted as a specific variant of ensemble copula coupling, and can thus be seen as belonging to this group of methods.





to the Appendix. Figures with additional results are available in the Supplementary Material. R (R Core Team, 2019) code with replication material and implementations of all methods is available from https://github.com/slerch/multiv_pp.

## 2 Post-processing of ensemble forecasts

We focus on multivariate ensemble post-processing approaches which are based on a combination of univariate post-processing models with copulas. The general two-step strategy of these methods is to first apply univariate post-processing to the ensemble forecasts for each margin (i.e., weather variable, location, and prediction horizon) separately. Then, in a second step, a suitably chosen copula is applied to the univariately post-processed forecasts in order to obtain the desired multivariate post-processing taking account of dependence patterns.

A copula is a multivariate cumulative distribution function (CDF) with standard uniform univariate marginal distributions
(Nelsen, 2006). The underlying theoretical background of the above procedure is given by Sklar's theorem (Sklar, 1959), which states that a multivariate CDF $H$ (this is what we desire) can be decomposed into a copula function $C$ modeling the dependence structures (this is what needs to be specified) and its marginal univariate CDFs $F_1, \ldots, F_d$ (this is what is obtained by the univariate post-processing) as follows:

$$H(x_1, \ldots, x_d) = C(F_1(x_1), \ldots, F_d(x_d))$$

for $x_1, \ldots, x_d \in \mathbb{R}$. In the approaches considered here, the copula $C$ is chosen to be either the non-parametric empirical copula induced by a pre-specified dependence template (in the ensemble copula coupling method and variants thereof as well as in the Schaake shuffle), or the parametric Gaussian copula (in the Gaussian copula approach, GCA). A Gaussian copula is a particularly convenient parametric model, as apart from the marginal distributions it only requires estimation of the correlation matrix of the multivariate distribution. Under a Gaussian copula the multivariate CDF $H$ takes the form

$$H(x_1, \ldots, x_d \,|\, \boldsymbol{\Sigma}) = \Phi_d\big(\Phi^{-1}(F_1(x_1)), \ldots, \Phi^{-1}(F_d(x_d)) \,|\, \boldsymbol{\Sigma}\big), \tag{1}$$

with $\Phi_d(\cdot \,|\, \boldsymbol{\Sigma})$ denoting the CDF of a $d$-dimensional standard normal distribution with mean zero and correlation matrix $\boldsymbol{\Sigma}$, and $\Phi^{-1}$ denoting the quantile function of the univariate standard normal distribution.

To describe the considered methods in more detail in what follows, let $\mathbf{X}_1, \ldots, \mathbf{X}_m \in \mathbb{R}^d$ denote unprocessed ensemble forecasts, where $\mathbf{X}_i := (X_i^{(1)}, \ldots, X_i^{(d)})$ for $i = 1, \ldots, m$, and let $\mathbf{y} := (y^{(1)}, \ldots, y^{(d)}) \in \mathbb{R}^d$ be the corresponding verifying
observation. We will use $l = 1, \ldots, d$ to denote a multi-index summarizing a fixed weather variable, location, and prediction horizon.

### 2.1 Step 1: Univariate post-processing

In a first step, univariate post-processing methods are applied to each margin $l = 1, \ldots, d$ separately. Prominent state-of-the-art univariate post-processing approaches include Bayesian model averaging (Raftery et al., 2005) and ensemble model output
statistics (EMOS; Gneiting et al., 2005). In the EMOS approach, which is employed throughout this paper, a non-homogeneous



distributional regression model

$$y^{(l)}|X_1^{(l)},\ldots,X_m^{(l)} \sim F_\theta^{(l)}(y^{(l)}|\boldsymbol{\theta}^{(l)})$$

is fitted, where $F_\theta^{(l)}$ is a suitably chosen parametric distribution with parameters $\boldsymbol{\theta}^{(l)} := g(X_1^{(l)},\ldots,X_m^{(l)})$ that depend on the unprocessed ensemble forecast through a link function $g(\cdot)$.

The choice of $F_\theta^{(l)}$ is in practice mainly determined by the weather variable being considered in the margin $l$. For instance, when $F_\theta^{(l)}$ can be assumed to be Gaussian with mean $\mu$ and variance $\sigma^2$, such as for temperature or pressure, one may set

$$F_\theta^{(l)} = \mathcal{N}(\mu,\sigma^2), \quad \text{where } (\mu,\sigma^2) := (a_0 + a_1\bar{X}, b_0 + b_1 S^2) = g(X_1^{(l)},\ldots,X_m^{(l)}) \tag{2}$$

if the ensemble members are exchangeable, with $\bar{X}$ and $S^2$ denoting the empirical mean and variance of the ensemble predictions $X_1^{(l)},\ldots,X_m^{(l)}$, respectively. The coefficients $a_0, a_1, b_0$ and $b_1$ are then derived via suitable estimation techniques using training data consisting of past ensemble forecasts and observations (Gneiting et al., 2005).

## 2.2 Step 2: Incorporating dependence structures using copulas to obtain multivariate post-processing

When applying univariate post-processing for each margin separately, multivariate (i.e. inter-variable, spatial and/or temporal) dependencies across the margins are lost. These dependencies are restored in a second step. Here, we consider five different approaches to do so.

**Assumption of independence (EMOS-Q)**

Instead of modeling the desired dependencies in any way, omitting the second step corresponds to assuming independence across the margins. To that end, a univariate sample $\hat{x}_1^{(l)},\ldots,\hat{x}_m^{(l)}$ is generated in each margin by drawing from the post-processed forecast distribution $F_\theta^{(l)}, l = 1,\ldots,d$. The univariate samples are then simply combined into a corresponding vector. Following Schefzik et al. (2013), we use equidistant quantiles of $F_\theta^{(l)}$ at levels $\frac{1}{m+1},\ldots,\frac{m}{m+1}$ to generate the sample, and denote this approach by EMOS-Q.

**Ensemble copula coupling (ECC)**

The basic ensemble copula coupling (ECC) approach proposed by Schefzik et al. (2013) proceeds as follows:

1. A sample $\hat{x}_1^{(l)},\ldots,\hat{x}_m^{(l)}$, where we assume $\hat{x}_1^{(l)} \leq \cdots \leq \hat{x}_m^{(l)}$ to simplify notation, of the same size $m$ as the unprocessed ensemble is drawn from each post-processed predictive marginal distribution $F_\theta^{(l)}, l = 1,\ldots,d$.

2. The sampled values are rearranged in the rank order structure of the raw ensemble, i.e., the permutation $\sigma_l$ of the set $\{1,\ldots,m\}$ defined by $\sigma_l(i) = \text{rank}(X_i^{(l)})$, with possible ties resolved at random, is applied to the post-processed sample from the first step in order to obtain the final ECC ensemble $\tilde{X}_1^{(l)},\ldots,\tilde{X}_m^{(l)}$ via

   $$\tilde{X}_i^{(l)} = \hat{x}_{\sigma_l(i)}^{(l)},$$

   where $i = 1,\ldots,m$ and $l = 1,\ldots,d$.





Depending on the specific sampling procedure in step 1, we here distinguish the following different ECC variants:

- **ECC-R**: The sample $\hat{x}_1^{(l)}, \ldots, \hat{x}_m^{(l)}$ is randomly drawn from $F_\theta^{(l)}$ (and subsequently arranged in ascending order).

- **ECC-Q**: The sample is constructed using equidistant quantiles of $F_\theta^{(l)}$ at levels $\frac{1}{m+1}, \ldots, \frac{m}{m+1}$:

$$\hat{x}_1^{(l)} := (F_\theta^{(l)})^{-1}\left(\frac{1}{m+1}\right), \ldots, \hat{x}_m^{(l)} := (F_\theta^{(l)})^{-1}\left(\frac{m}{m+1}\right).$$

- **ECC-S** (Hu et al., 2016): First, random numbers $u_1, \ldots, u_m$, where $u_i \sim \mathcal{U}(\frac{i-1}{m}, \frac{i}{m}]$ for $i = 1, \ldots, m$, are drawn, with
$\mathcal{U}(a, b]$ denoting the uniform distribution on the interval $(a, b]$. Then, $\hat{x}_i^{(l)}$ is set to the quantile of $F_\theta^{(l)}$ at level $u_i$:

$$\hat{x}_1^{(l)} := (F_\theta^{(l)})^{-1}(u_1), \ldots, \hat{x}_m^{(l)} := (F_\theta^{(l)})^{-1}(u_m).$$

Besides the above sampling schemes, Schefzik et al. (2013) propose an alternative transformation approach referred to as ECC-T. This variant is in particular appealing for theoretical considerations, as it provides a link between the ECC notion and member-by-member post-processing approaches (Schefzik, 2017). However, as it may involve additional modeling steps,
ECC-T is not as generic as the other schemes and thus not explicitly considered in this paper.

**Dual ensemble copula coupling (dECC)**

Dual ECC (dECC) is an extension of ECC which aims at combining the structure of the unprocessed ensemble with a component accounting for the forecast error autocorrelation structure (Ben Bouallègue et al., 2016), proceeding as follows:

1. ECC-Q is applied in order to obtain re-ordered ensemble forecasts $\tilde{\mathbf{X}}_1, \ldots, \tilde{\mathbf{X}}_m$, with $\tilde{\mathbf{X}}_i := (\tilde{X}_i^{(1)}, \ldots, \tilde{X}_i^{(d)})$ for $i =$
$1, \ldots, m$.

2. A transformation based on an estimate of the error autocorrelation $\hat{\mathbf{\Sigma}}_e$ is applied to the bias-corrected post-processed forecast in order to obtain correction terms $\mathbf{c}_1, \ldots, \mathbf{c}_m$. Precisely, $\mathbf{c}_i := (\hat{\mathbf{\Sigma}}_e)^{\frac{1}{2}} \cdot (\tilde{\mathbf{X}}_i - \mathbf{X}_i)$ for $i = 1, \ldots, m$.

3. An adjusted ensemble $\breve{\mathbf{X}}_1, \ldots, \breve{\mathbf{X}}_m$ is derived via $\breve{\mathbf{X}}_i := \mathbf{X}_i + \mathbf{c}_i$. for $i = 1, \ldots, m$.

4. ECC-Q is applied again, but now performing the re-ordering with respect to the rank order structure of the adjusted
ensemble from step 3 used as a modified dependence template.

**Schaake shuffle (SSh)**

The Schaake shuffle (SSh) proceeds as ECC-Q, but re-orders the sampled values in the rank order structure of $m$ randomly determined past observations (Clark et al., 2004) and not with respect to the unprocessed ensemble forecasts. For a better comparison with (d)ECC, the size of the SSh ensemble is restricted to equal that of the unprocessed ensemble here. However,
in principle, the SSh ensemble may have an arbitrary size, provided that sufficiently enough past observations are available to build the dependence template. Extensions of the SSh that select past observations based on similarity are available (Schefzik, 2016; Scheuerer et al., 2017), but not explicitly considered here as their implementation is not directly straightforward and may involve additional modelling choices specific to the situation at hand.





The reordering-based methods considered thus far can be interpreted as non-parametric, empirical copula approaches. In particular, in the setting of Sklar's theorem, $C$ is taken to be the empirical copula induced by the corresponding dependence template, i.e., the unprocessed ensemble forecasts in case of ECC, the adjusted ensemble in case of dECC, and the past observations in case of the SSh.

**Gaussian copula approach (GCA)**

By contrast, in the Gaussian copula approach (GCA) proposed by Pinson and Girard (2012) and Möller et al. (2013), the copula $C$ is taken to be the parametric Gaussian copula. GCA can be traced back to similar ideas from earlier work in spatial statistics (e.g., Berrocal et al., 2008) and proceeds as follows:

1. A set of past observations $\mathbf{y}_1, \ldots, \mathbf{y}_K$, with $\mathbf{y}_k = (y_k^{(1)}, \ldots, y_k^{(d)})$, is transformed into latent standard Gaussian observations $\tilde{\mathbf{y}}_1, \ldots, \tilde{\mathbf{y}}_K$ by setting

$$\tilde{y}_k^{(l)} = \Phi^{-1}\left(F_\theta^{(l)}(y_k^{(l)})\right) \tag{3}$$

   for $k = 1, \ldots, K$ and $l = 1, \ldots, d$, where $\Phi^{-1}$ is the inverse of the CDF of the univariate standard normal distribution and $F_\theta^{(l)}$ is the marginal distribution obtained by univariate post-processing. The index $k = 1, \ldots, K$ here refers to a training set of past observations.

2. An empirical (or parametric) $(d \times d)$ correlation matrix $\widehat{\boldsymbol{\Sigma}}$ of the $d$-dimensional standard normal distribution in (1) is estimated from $\tilde{\mathbf{y}}_1, \ldots, \tilde{\mathbf{y}}_K$.

3. Multivariate random samples $\mathbf{Z}_1, \ldots, \mathbf{Z}_m \sim \mathcal{N}_d(\mathbf{0}, \widehat{\boldsymbol{\Sigma}})$ are drawn, where $\mathcal{N}_d(\mathbf{0}, \widehat{\boldsymbol{\Sigma}})$ denotes a $d$-dimensional normal distribution with mean vector $\mathbf{0} := (0, \ldots, 0)$ and estimated correlation matrix $\widehat{\boldsymbol{\Sigma}}$ from Step 2, and $\mathbf{Z}_i := (Z_i^{(1)}, \ldots, Z_i^{(d)})$ for $i = 1, \ldots, m$.

4. The final GCA post-processed ensemble forecast $\mathbf{X}_1^*, \ldots, \mathbf{X}_m^*$, with $\mathbf{X}_i^* := (X_i^{*(1)}, \ldots, X_i^{*(d)})$ for $i = 1, \ldots, m$ is obtained via

$$X_i^{*(l)} := \left(F_\theta^{(l)}\right)^{-1}\left(\Phi(Z_i^{(l)})\right) \tag{4}$$

   for $i = 1, \ldots, m$ and $l = 1, \ldots, d$, with $\Phi$ denoting the CDF of the univariate standard normal distribution.

## 3 Simulation settings

We consider several simulation settings to highlight different aspects and provide a broad comparison of the effects of potential misspecifications of the ensemble predictions on the performance of the various multivariate post-processing methods. The general setup of all simulation settings is as follows.





An initial training set of pairs of simulated ensemble forecasts and observations of size $n_{\text{init}}$ is generated. Post-processed forecasts are then computed and evaluated over a test set of size $n_{\text{test}}$. Therefore, $n := n_{\text{init}} + n_{\text{test}}$ iterations are performed in total for all simulation settings. In the following, we set $m = 50, n_{\text{init}} = 500, n_{\text{test}} = 1\,000$ throughout.

To describe the individual settings in more detail, we here begin by first identifying the general structure of the steps that
are performed in all settings. For each iteration $t$ in both training and test set (i.e., $t = 1, \ldots, n$), multivariate forecasts and observations are generated:

(S1) Generate multivariate observations and ensemble forecasts.

For all iterations $t$ in the test set (i.e., $t = n_{\text{init}} + 1, \ldots, n$), the following steps are carried out:

(S2) Apply univariate post-processing separately in each dimension.[2]

(S3) Apply multivariate post-processing methods.

(S4) Compute univariate and multivariate measures of forecast performance on the test set.

Unless indicated otherwise all simulation draws are independent across iterations. To simplify notation we will thus typically omit the simulation iteration index $t$ in the following.

To quantify simulation uncertainty, the above procedure is repeated 100 times for each tuning parameter combination in each setting. In the interest of brevity, we omit ECC-R which did show substantially worse results in initial tests (see also Schefzik et al., 2013). In the following, the individual simulation settings are described in detail, and specific implementation choices are discussed.

### 3.1 Setting 1: Multivariate Gaussian distribution

As starting point we first consider a simulation model where observations and ensemble forecasts are drawn from multivariate Gaussian distributions.[3] The simplicity of this model allows to readily interpret misspecifications in the mean, variance and covariance structures.

(S1) For iterations $t = 1, \ldots, n$, independent and identically distributed samples of observations and ensemble forecasts are generated as follows:

- observation: $\mathbf{y} \sim \mathcal{N}_d(\boldsymbol{\mu_0}, \boldsymbol{\Sigma^0})$, where $\boldsymbol{\mu_0} = (0, \ldots, 0) \in \mathbb{R}^d$, and $\Sigma_{i,j}^0 = \rho_0^{|i-j|}$, for $i, j = 1, \ldots, d$.

- ensemble forecasts: $\mathbf{X}_1, \ldots, \mathbf{X}_m \overset{\text{iid}}{\sim} \mathcal{N}_d(\boldsymbol{\mu}, \boldsymbol{\Sigma})$, where $\boldsymbol{\mu} = (\epsilon, \ldots, \epsilon) \in \mathbb{R}^d$, and $\Sigma_{i,j} = \sigma\,\rho^{|i-j|}$, for $i, j = 1, \ldots, d$.

---

[2]With the exception of Setting 4, the estimation of univariate post-processing models utilizes the initial training set only. Setting 4 covers the possibly more realistic case of variations across repetitions of the experiment.

[3]Wilks (2017) considers a similar setting in the context of multivariate calibration assessment which we here extend towards multivariate ensemble post-processing.





The parameters $\epsilon$ and $\sigma$ introduce a bias and a misspecified variance in the marginal distributions of the ensemble forecasts. These systematic errors are kept constant across dimensions $1, \ldots, d$. The parameters $\rho_0$ and $\rho$ control the autoregressive structure of the correlation matrix of the observations and ensemble forecasts. Setting $\rho_0 \neq \rho$ introduces misspecifications of the correlation structure of the ensemble forecasts.

(S2) As described in Section 2.1, univariate post-processing is applied independently in each dimension $1, \ldots, d$. Here, we employ the standard Gaussian EMOS model (2) proposed by Gneiting et al. (2005). The EMOS coefficients $a_0, a_1, b_0, b_1$ are estimated by minimizing the mean continuous ranked probability score (CRPS, see Appendix B) over the training set consisting of the $n_{\text{init}}$ initial iterations, and are then used to produce out of sample forecasts for the $n_{\text{test}}$ iterations in the test set.

(S3) Next, the multivariate post-processing methods described in Section 2.2 are applied. Implementation details for the individual methods are as follows.

- For dECC, the estimate of the error autocorrelation $\hat{\boldsymbol{\Sigma}}_e$ is obtained from the $n_{\text{init}}$ initial training iterations to compute the required correction terms for the test set.

- To obtain the dependence template for SSh, $m$ past observations are randomly selected from all iterations preceding the current iteration $t$.

- The correlation matrix $\boldsymbol{\Sigma}$ required for GCA is estimated by the empirical correlation matrix based on all iterations preceding the current iteration $t$.

- The verification results for all methods that require random sampling (ECC-S, SSh, GCA) are averaged over 10 independent repetitions for each iteration $t = n_{\text{init}} + 1, \ldots, n$ in the test set.

The multivariate Gaussian setting is implemented for $d = 5$ and all combinations of $\epsilon \in \{0, 1, 3\}, \sigma^2 \in \{0.5, 1, 2, 5\}$, and $\rho, \rho_0 \in \{0.1, 0.25, 0.5, 0.75, 0.9\}$. As indicated above, the simulation experiment is repeated 100 times for each of the 300 parameter combinations.

## 3.2 Setting 2: Multivariate truncated Gaussian distribution

Setting 1 can be generalized by replacing the multivariate Gaussian distribution by a multivariate truncated Gaussian distri-
bution $\mathcal{N}_{\boldsymbol{a},\boldsymbol{b}}^d(\boldsymbol{\mu}, \boldsymbol{\Sigma})$, where $\boldsymbol{a}$ and $\boldsymbol{b}$ are the vectors of lower and upper truncation points, respectively. In univariate settings this distribution plays important role in wind speed modelling (Thorarinsdottir and Gneiting, 2010) or in post-processing of hydrological forecasts (Hemri and Klein, 2017). Compared to Setting 1, here misspecifications in location vector $\boldsymbol{\mu}$ and/or scale matrix $\boldsymbol{\Sigma}$ result in more complex deviations in mean vectors and covariance matrices.

(S1) For iterations $t = 1, \ldots, n$, independent and identically distributed samples of observations $\mathbf{y}$ and ensemble forecasts
$\mathbf{X}_1, \ldots, \mathbf{X}_m$ are generated from $\mathcal{N}_{\boldsymbol{a},\boldsymbol{b}}^d(\boldsymbol{\mu_0}, \boldsymbol{\Sigma^0})$ and $\mathcal{N}_{\boldsymbol{a},\boldsymbol{b}}^d(\boldsymbol{\mu}, \boldsymbol{\Sigma})$, respectively, where $\boldsymbol{\Sigma^0}$ and $\boldsymbol{\Sigma}$ are defined as in Setting 1, $\boldsymbol{\mu_0} = (\mu_0, \ldots, \mu_0) \in \mathbb{R}^d$ and $\boldsymbol{\mu} = (\mu, \ldots, \mu) \in \mathbb{R}^d$.





(S2) Univariate post-processing is based on the truncated normal EMOS model of Hemri and Klein (2017), where the EMOS coefficients are calculated by optimizing the mean CRPS over the training set consisting of the $n_{\mathrm{init}}$ initial iterations. Similar to Setting 1, the obtained EMOS models are used to produce out of sample forecasts for the $n_{\mathrm{test}}$ iterations in the test set.


(S3) Identical to (S3) of Setting 1.

For simplicity, we consider a lower truncation at 0 only, i.e., $\boldsymbol{a} = (0, \ldots, 0)$ and $\boldsymbol{b} = (\infty, \ldots, \infty)$. The truncated Gaussian setting is implemented for $d = 5$ and all combinations of

$$\mu_0 \in \{2,3\}, \ \mu \in \{2,3,5\}, \ \rho_0 \in \{0.25, 0.5, 0.75\}, \ \rho \in \{0.1, 0.25, 0.5, 0.75, 0.9\}, \ \sigma \in \{0.25, 0.5, 1, 3, 5\},$$

resulting in 450 experiments which are repeated 100 times each.

### 3.3 Setting 3: Multivariate censored extreme value distribution

To investigate alternative marginal distributions employed in post-processing applications, we further consider a simulation
setting based on a censored version of the generalized extreme value (GEV) distribution. The GEV distribution was introduced by Jenkinson (1955) among others, combining three different types of extreme value distributions. It has been widely used for modelling extremal climatological events such as flood peaks (e.g., Morrison and Smith, 2002) or extreme precipitation (e.g., Feng et al., 2007). In the context of post-processing, GEV distributions have for example been applied for modeling wind speed in Lerch and Thorarinsdottir (2013). Here, we consider multivariate observations and forecasts with marginal
distributions given by a left-censored version of the GEV distribution which was proposed by Scheuerer (2014) in the context of post-processing ensemble forecasts of precipitation amounts.

(S1) For iterations $t = 1, \ldots, n$ samples of observations and ensemble forecasts are generated as follows. For $l = 1, \ldots, d$, the marginal distributions are GEV distributions left-censored at 0,

$$F_\theta^{(l)} = \mathrm{GEV}_0(\mu, \sigma, \xi),$$

where the distribution parameters $\mu$ (location), $\sigma$ (scale) and $\xi$ (shape) are identical across dimensions $l = 1, \ldots, d$. Details on the left-censored GEV distribution are provided in Appendix A. Misspecifications of the marginal ensemble predictions are obtained by choosing different GEV parameters for observations $(\mu_y, \sigma_y, \xi_y)$ and forecasts $(\mu_x, \sigma_x, \xi_x)$. Combined misspecifications of the three parameters result in more complex deviations of mean and variance (on the univariate level) especially compared to Setting 1, but also compared to Setting 2. Typically there is a joint influence of
the GEV parameters on mean and dispersion properties of the distribution. In order to exploit the complex behavior a variety of parameter combinations for observations and ensemble forecasts were considered.

To generate multivariate observations $\mathbf{y} = (y^{(1)}, \ldots, y^{(d)})$ and ensemble predictions $\mathbf{X}_i = (X_i^{(1)}, \ldots, X_i^{(d)})$, $i = 1, \ldots, m$, the so-called NORTA (normal to anything) approach is chosen, see Cario and Nelson (1997); Chen (2001). This method




allows to generate realizations of a random vector $\boldsymbol{z} = (z^{(1)}, \ldots, z^{(d)})$ with specified marginal distribution functions $F_\theta^{(l)}, l = 1, \ldots, d$, and a given correlation matrix $R = (\mathrm{Corr}(z^{(k)}, z^{(l)}))_{k,l=1}^d$.

The NORTA procedure consists of three steps. In a first step a vector $\boldsymbol{v} = (v^{(1)}, \ldots, v^{(d)})$ is generated from $\mathcal{N}_d(0, R^*)$ for a correlation matrix $R^*$. In a second step, $u^{(l)} = \Phi(v^{(l)})$ is computed, where $\Phi$ denotes the CDF of the standard normal distribution. In a third step, $z^{(l)} = \left(F_\theta^{(l)}\right)^{-1}(u^{(l)})$ is derived for $l = 1, \ldots, d$, where $\left(F_\theta^{(l)}\right)^{-1}$ is the inverse of $F_\theta^{(l)}$. The correlation matrix $R^*$ is chosen in a such a way that the $z^{(l)}$ have the desired target correlation matrix $R$. Naturally, the specification of $R^*$ is the most involved part of this procedure. Here, we use the retrospective approximation algorithm implemented in the `R` package `NORTARA` (Su, 2014). The `NORTARA` package infrequently produced error and warnings, which were not present for alternative starting values of the random number generator.

Following the previous simulation settings the target correlation matrix $R$ is chosen as

$$R_{i,j} = \rho^{|i-j|}$$

for $-1 < \rho < 1$ and $i, j = 1, \ldots, d$.

(S2) To separately post-process the univariate ensemble forecasts we employ the EMOS method for quantitative precipitation based on the left-censored GEV distribution proposed by Scheuerer (2014). To that end we assume $-0.278 < \xi < 0.5$, in which case the mean $\nu$ and the variance of the non-censored GEV exist, and

$$\nu = \begin{cases} \mu + \sigma \frac{\Gamma(1-\xi)-1}{\xi}, & \xi \neq 0 \\ \mu + \sigma\gamma, & \xi = 0 \end{cases},$$

where $\Gamma$ denotes the gamma function and $\gamma$ is the Euler-Mascheroni constant. See Appendix A for comments on mean and variance of the left-censored GEV. Following Scheuerer (2014), the parameters $(\nu, \sigma, \xi)$ are linked to the ensemble predictions via

$$g(X_1^{(l)}, \ldots, X_m^{(l)}) = \left(a_0 + a_1 \bar{X}^{(l)} + a_2 \bar{X}_0^{(l)}, b_0 + b_1 \mathrm{MD}_X^{(l)}, \xi\right).$$

Here, $\bar{X}^{(l)}$ and $\bar{X}_0^{(l)}$ are the arithmetic mean and the fraction of zero values of the ensemble predictions $X_1^{(l)}, \ldots, X_m^{(l)}$, respectively, while $\mathrm{MD}_X^{(l)}$ denotes the mean absolute difference of the ensemble predictions, i.e.,

$$\mathrm{MD}_X^{(l)} = \frac{1}{m^2} \sum_{i=1}^m \sum_{j=1}^m \left| X_i^{(l)} - X_j^{(l)} \right|.$$

The shape parameter $\xi$ is not linked to the ensemble predictions, but is estimated along with the EMOS coefficients $a_0$, $a_1$, $a_2$ and $b_0$, $b_1$. As in Scheuerer (2014), the link function refers to the parameter $\nu$ instead of $\mu$, since it is argued that for fixed $\nu$ an increase in $\sigma$ can be interpreted more naturally as an increase in uncertainty.

An implementation in `R` is available in the `ensembleMOS` package (Yuen et al., 2018). For our simulation, this package was not directly invoked, but the respective functions were used as a template. As described in Section 2.1, univariate





post-processing is applied independently in each dimension $l = 1, \ldots, d$. The EMOS coefficients are estimated as described above over the training set consisting of the $n_{\text{init}}$ initial iterations, and are then used to produce out of sample forecasts for the $n_{\text{test}}$ iterations in the test set.

(S3) Identical to (S3) of Setting 1, except for GCA, where we proceed differently to account for the point mass at zero. The latent standard Gaussian observations $\tilde{y}_k^{(l)}$ are generated by $\tilde{y}_k^{(l)} = \Phi^{-1}(u)$, where $u$ is a randomly chosen value in the interval $(0, F_\theta^{(l)}(0))$ in case $y_k^{(l)} = 0$ and $u = F_\theta^{(l)}(y_k^{(l)})$ in case $y_k^{(l)} > 0$.

|   | $\mu_y$ | $\xi_y$ | $\sigma_y$ | $\mu_x$ | $\xi_x$ | $\sigma_x$ |
|---|---|---|---|---|---|---|
| A | 0.0 | -0.1 | 1.0 | 1.0 | 0.0 | 0.2 |
| B | 0.0 | -0.1 | 1.0 | 0.0 | 0.0 | 2.0 |
| C | 1.0 | 0.3 | 1.0 | 0.0 | 0.0 | 2.0 |
| D | 0.0 | 0.0 | 1.0 | 0.0 | 0.0 | 1.0 |

**Table 1.** Different simulation scenarios for Setting 3.

The multivariate censored extreme value setting is implemented for $d = 4$ and four different scenarios summarized in Table 1. In each scenario the $\text{GEV}_0$ distribution parameters for the observations are chosen according to $(\mu_y, \xi_y, \sigma_y)$, while the parameters for the ensemble predictions are chosen according to $(\mu_x, \xi_x, \sigma_x)$. In both cases, the correlation matrix $R$ from above is invoked with different choices of $\rho_y$ and $\rho_x$ from the set $\{0.25, 0.5, 0.75\}$ giving a total of $4 \times 9 = 36$ scenarios.
Note that according to Scheuerer (2014) there is a positive probability for zero to occur when either $\xi \leq 0$ or $\xi > 0$ and $\mu < \sigma/\xi$. The scenarios from Table 1 are chosen in such a way that either one of these two conditions is met.

### 3.4 Setting 4: Multivariate Gaussian distribution with changes over iterations

In the preceding simulation settings, the misspecifications of the ensemble forecasts were kept constant over the iterations $t = 1, \ldots, n$ within the simulation experiments. However, forecast errors of real-world ensemble predictions often exhibit systematic changes over time, for example due to seasonal effects or differences in flow-dependent predictability due to variations of large scale atmospheric conditions. Here, we modify the multivariate Gaussian simulation setting from Section 3.1 to introduce changes in the mean, variance and covariance structure of the multivariate distributions of observations and ensemble forecasts. In analogy to practical applications of multivariate post-processing, the ensemble predictions and observations may be interpreted as multivariate in terms of location or prediction horizon, with changes of the misspecification properties over time.

(S1) For iterations $t = 1, \ldots, n$, independent samples of observations and ensemble forecasts are generated as follows:





 – observation: $\mathbf{y} \sim \mathcal{N}_d(\boldsymbol{\mu_0}, \boldsymbol{\Sigma^0})$, where $\boldsymbol{\mu_0} = \sin\left(\frac{2\pi t}{n}\right) + (\epsilon_0, \ldots, \epsilon_0)^T \in \mathbb{R}^d$. To obtain the correlation matrix $\boldsymbol{\Sigma^0}$, let $R_{i,j} = \rho_0^{|i-j|} + \sin\left(\frac{2\pi t}{n}\right)$, for $i, j = 1, \ldots, d$ and $S_0 = RR^T$. The covariance matrix $S_0$ is scaled into the corresponding correlation matrix $\boldsymbol{\Sigma^0}$ using the R function `cov2cor()`.

295 – ensemble forecasts: $\mathbf{X}_1, \ldots, \mathbf{X}_m \overset{\text{iid}}{\sim} \mathcal{N}_d(\boldsymbol{\mu}, \boldsymbol{\Sigma})$, where $\boldsymbol{\mu} = \sin\left(\frac{2\pi t}{n}\right) + (\epsilon, \ldots, \epsilon)^T \in \mathbb{R}^d$. To obtain the correlation matrix $\boldsymbol{\Sigma}$ we proceed as for the observations, however, we set $R_{i,j} = \rho^{|i-j|} + \sin\left(\frac{2\pi t}{n}\right)$, for $i, j = 1, \ldots, d$ (i.e., $\rho_0$ is replaced by $\rho$).

In contrast to Setting 1, the misspecifications in the mean and correlation structure now include a periodic component.

(S2) As in Setting 1, we employ the standard Gaussian EMOS model (2). However, to account for the changes over iterations we now utilize a rolling window consisting of pairs of ensemble forecasts and observations from the 100 iterations preceding $t$ as training set to obtain estimates of the EMOS coefficients. See Lang et al. (2019a) for a detailed discussion of alternative approaches to incorporate time dependence in the estimation of post-processing models.

(S3) The application of the multivariate post-processing methods is identical to the approach taken in Setting 1. Note that we deliberately follow the naive standard implementations (see Section 2.2) here to highlight some potential issues of the Schaake shuffle in this context.

The above setting is implemented for $d = 5, \epsilon = 1, \sigma^2 = 1$ and all combinations of $\rho, \rho_0 \in \{0.1, 0.25, 0.5, 0.75, 0.9\}$. As before the simulation experiment is repeated 100 times for each of the parameter combinations.

## 4   Results

In the following, we focus on comparisons of the relative predictive performance of the different multivariate post-processing methods and apply proper scoring rules for forecast evaluation. In particular, we use the energy score (ES; Gneiting et al., 2008) and variogram score of order 1 (VS; Scheuerer and Hamill, 2015) to evaluate multivariate forecast performance. Diebold-Mariano (DM; Diebold and Mariano, 1995) tests are applied to assess the statistical significance of the score differences between models. Details on forecast evaluation based on proper scoring rules and DM tests are provided in Appendix B. Note that proper scoring rules are often used in the form of skill scores to investigate relative improvements in predictive performance in the meteorological literature. Here, we instead follow suggestions of Ziel and Berk (2019) who argue that the use of DM tests is of crucial importance to appropriately discriminate between multivariate models.

While our focus here is on multivariate performance, we briefly demonstrate that the univariate post-processing models applied in the different simulation settings usually work as intended.

### 4.1   Univariate performance

The univariate predictive performance of the raw ensemble forecasts in terms of the CRPS is improved by the application of univariate post-processing methods across all parameter choices in all simulation settings. The magnitude of the relative


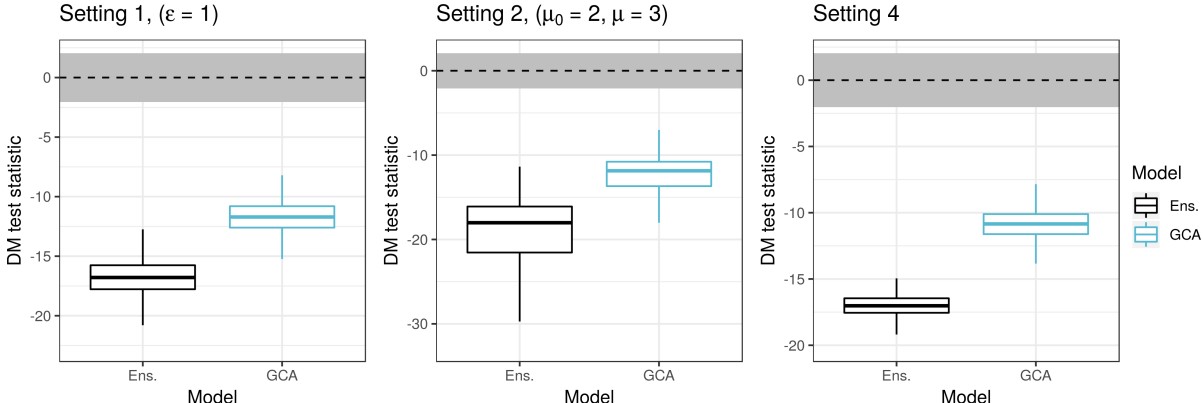

**Figure 1.** Summaries of DM test statistic values based on the CRPS. ECC-Q forecasts are used as reference model such that positive values of the test statistic indicate improvements over ECC-Q and negative value indicated deterioration of forecast skill. Boxplots summarize results of from multiple parameter combinations for the simulation settings, with restrictions on the simulation parameters indicated in the plot title. For example, boxplots in the first panel summarize simulation results from all parameter combinations of Setting 1 (and the 100 Monte Carlo repetitions each) subject to $\epsilon = 1$. The horizontal gray stripe indicates the acceptance region of the two-sided DM test under the null hypothesis of equal predictive performance at a level of 0.05.

improvements by post-processing depends on the chosen simulation parameters, exemplary results are shown in Figure 1. Note that ECC-S, dECC and SSh are omitted as the univariate forecast distributions are identical to those of ECC-Q, subject to random fluctuations due to the 10 random repetitions that were performed to account for simulation uncertainty of those

methods.[4]

For the simulation parameter values summarized there, univariate post-processing works as intended with statistically significant improvements over the raw ensemble forecasts. Note that for GCA the univariate marginal distributions are modified due to the transformation step in (4). While the quantile forecasts of ECC-Q are optimal in terms of the CRPS (Bröcker, 2012) the univariate GCA forecasts do not possess this property, resulting in worse univariate performance compared to all other

methods.

To give an impression of the univariate performance of raw ensemble forecasts compared to post-processing for Setting 3, the four scenarios from Table 1 are considered and CRPS skill scores for the raw ensemble are computed with ECC-Q as reference model, see Figure 2. Negative skill score values indicate an improvement of ECC-Q over the raw ensemble, while positive values indicate the contrary. Scenario D exhibits the smallest level of improvement, which is to expected, since scenario D

reflects a situation where raw ensemble forecasts stem from the same distribution as the observations.

---

[4]In fact, ECC-Q does not change the marginal distributions, the univariate forecasts are thus identical to solely applying univariate post-processing methods in the margins separately, without accounting for dependencies. We will later refer to this as EMOS-Q.

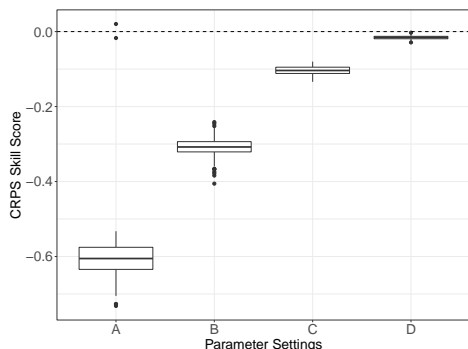

**Figure 2.** CRPS skill score of raw ensemble with ECC-Q as reference model for the four different scenarios considered in Setting 3. The boxplots summarize results over 100 repetitions of each individual experiment.

## 4.2 Multivariate performance

We now compare the multivariate performance of the different post-processing approaches presented in Section 2.2. Multivariate forecasts obtained by only applying the univariate post-processing methods without accounting for dependencies (denoted by EMOS-Q) as well as the raw ensemble predictions (ENS) are usually significantly worse and will be omitted in most com-
parisons below unless indicated otherwise. Additional figures with results for all parameter combinations in all settings are provided in the Supplementary Material.

### 4.2.1 Setting 1: Multivariate Gaussian distribution

The tuning parameter $\epsilon$ governing the bias in the mean vector of the ensemble forecasts only has very limited effects on the relative performance of the multivariate post-processing methods. To retain focus we restrict our attention to $\epsilon = 1$. Figure 3
shows results in terms of the ES for two different choices of $\sigma$, using multivariate forecasts of ECC-Q as reference method. For visual clarity, we omit parameter combinations where either $\rho \in \{0.1, 0.9\}$ or $\rho_0 \in \{0.1, 0.9\}$. Corresponding results are available in the Supplementary Material. Note that the relative forecast performance of all approaches except for dECC generally does not depend on $\sigma$, we thus proceed to discuss the remaining approaches first, and dECC last.

If the correlation structure of the unprocessed ensemble forecasts is correctly specified (i.e., $\rho = \rho_0$), no significant dif-
ferences can be detected between ECC-Q, ECC-S and SSh. In contrast, GCA (and dECC for larger values of $\sigma$) perform substantially worse. The worse performance of GCA might be due to the larger forecast errors in the univariate margins, see Section 4.1.

In the cases with misspecifications in the correlation structure (i.e., $\rho \neq \rho_0$), larger differences can be detected among all methods. Notably, SSh never performs substantially worse than ECC-Q and is always among the best performing approaches.
The larger the absolute difference between $\rho$ and $\rho_0$, the greater the improvement of SSh relative to ECC-Q as it becomes more and more beneficial to learn the dependence template from past observations rather than the raw ensemble the less information


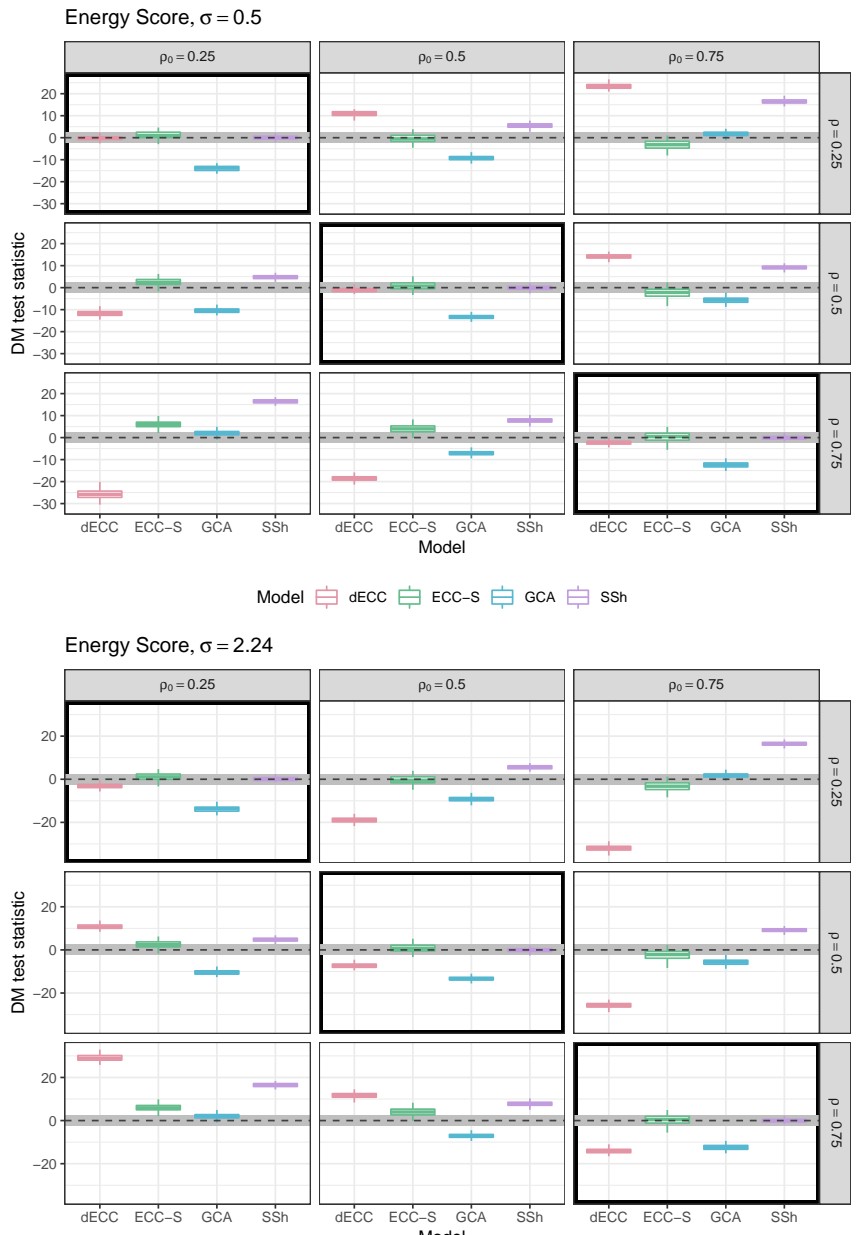

**Figure 3.** Summaries of DM test statistic values based on the ES for Setting 1 with $\epsilon = 1$, and $\sigma = 0.5$ (top), and $\sigma = \sqrt{5}$ (bottom). ECC-Q forecasts are used as reference model such that positive values of the test statistic indicate improvements over ECC-Q and negative value indicated deterioration of forecast skill. Boxplots summarize results of the 100 Monte Carlo repetitions of each individual experiment. The horizontal gray stripe indicates the acceptance region of the two-sided DM test under the null hypothesis of equal predictive performance at a level of 0.05. Simulation parameter choices where the correlation structure of the raw ensemble is correctly specified ($\rho = \rho_0$) are surrounded by black boxes.





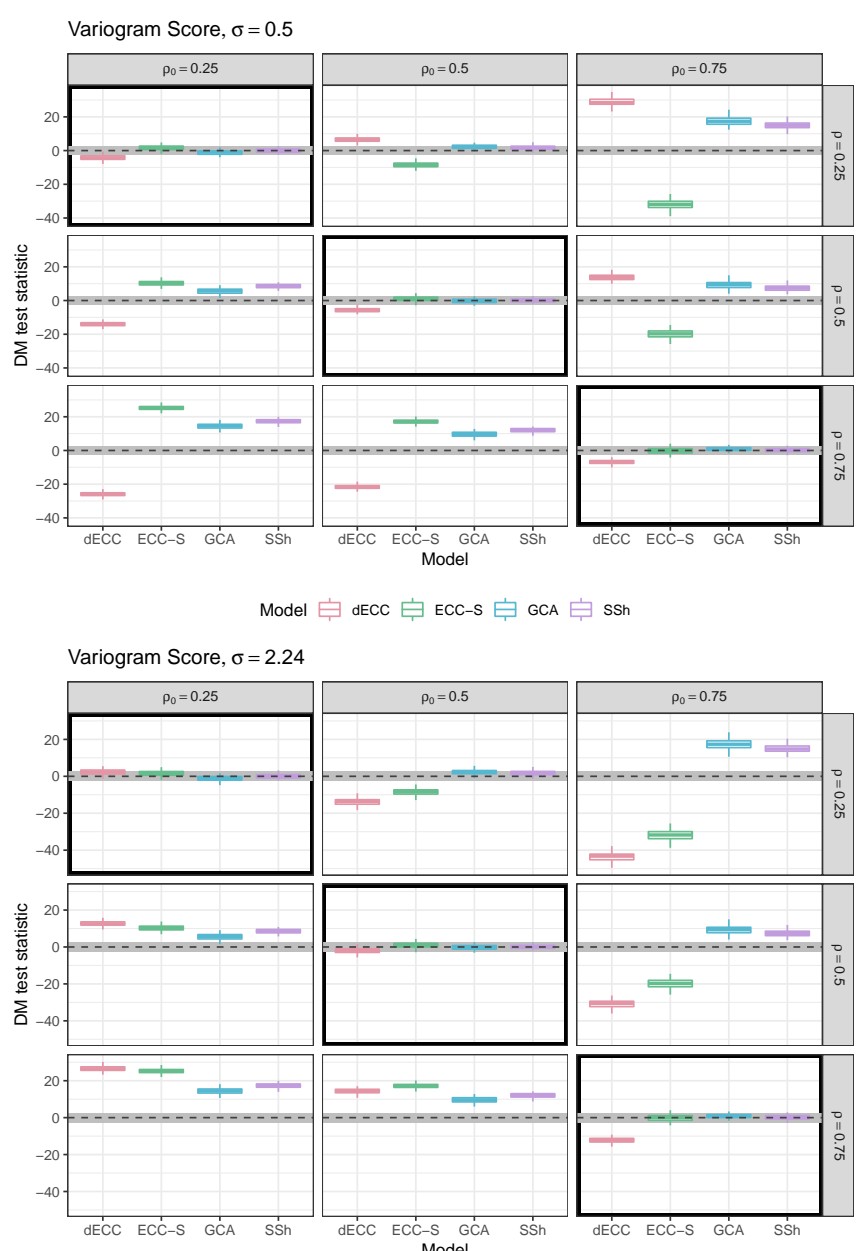

**Figure 4.** As Figure 3, but summarizing results in terms of the VS.





the ensemble provides about the true dependence structure. GCA also tends to outperform ECC-Q if the differences between $\rho$ and $\rho_0$ are large, however, GCA always performs worse than SSh and shows significantly worse performance than ECC-Q if the misspecifications in the ensemble are not too large (i.e., if $\rho$ and $\rho_0$ are equal or similar).

The relative performance of ECC-S depends on the ordering of $\rho$ and $\rho_0$. If $\rho > \rho_0$, ECC-S significantly outperforms ECC-Q, however, if $\rho < \rho_0$ significant ES differences in favor of ECC-Q can be detected. For dECC, the performance further depends on the misspecification of the variance structure in the marginal distributions. If $\rho > \rho_0$, the DM test statistic values move from positive (improvement over ECC-Q) to negative (deterioration compared to ECC-Q) values for increasing $\sigma$. By contrast, if $\rho < \rho_0$ the values of the test statistic instead change from negative to positive for increasing $\sigma$. The differences are mostly

statistically significant, and indicate the largest relative improvements among all methods in cases of the largest possible differences between $\rho$ and $\rho_0$. However, note that for some of those parameter combinations with small $\rho$ and large $\rho_0$, even EMOS-Q can outperform ECC-Q and ECC-S. In these situations, the raw ensemble forecasts contain very little information about the dependence structure and the ES can be improved by assuming independence instead of learning the dependence template from the ensemble.

Results in terms of the VS are shown in Figure 4. Most of the conclusions from the results in terms of the ES extend directly to comparisons based on the VS. SSh consistently remains among the best performing methods and provides significant improvements over ECC-Q unless $\rho = \rho_0$, however, alternative approaches here outperform SSh more often. Notably, the relative performance of GCA is consistently better in terms of the VS than in terms of the ES. For example, the differences between GCA and SSh appear to generally be negligible and GCA does not perform worse than ECC-Q for any of the simulation pa-

rameter combinations. These differences between the results for GCA in terms of ES and VS can potentially be explained by the greater sensitivity of the VS to misspecifications in the correlation structure, whereas the ES shows a stronger dependence on the mean vector.

For ECC-S and dECC, the general dependence on values of $\rho$, $\rho_0$ and $\sigma$ (for dECC) is similar to the results for the ES, but the magnitude of both positive as well as negative differences to all other methods is increased. For example, it is now possible

to find parameter combinations where either ECC-S or dECC (or both) substantially outperform both GCA and SSh.

### 4.2.2   Setting 2: Multivariate truncated Gaussian distributions

Figure 5 summarizes results for Setting 2 in terms of the ES (top) and the VS (bottom). In the interest of brevity, we only show results for $\sigma = 1$ and $\rho, \rho_0 \in \{0.25, 0.5, 0.75\}$, but discuss the effect of misspecifications of the variance below. Corresponding plots are provided in the Supplemental Material.

Overall, SSh consistently provides significant improvements over ECC-Q except for settings in which the correlation structure of the ensemble is correctly specified ($\rho = \rho_0$) where no significant differences can be detected. The relative differences in favor of SSh are increased for larger absolute differences of $\rho$ and $\rho_0$. While these findings are in line with the results from Setting 1 and hold for both ES and VS, the relative performance of GCA is now somewhat different from before. In particular terms of the ES, GCA here performs worse for many parameter settings. In particular, GCA is significantly worse than ECC-Q

if $\rho = \rho_0$, and does not offer any improvements over ECC-Q if $\rho$ and $\rho_0$ are not too different. By contrast, in terms of the VS



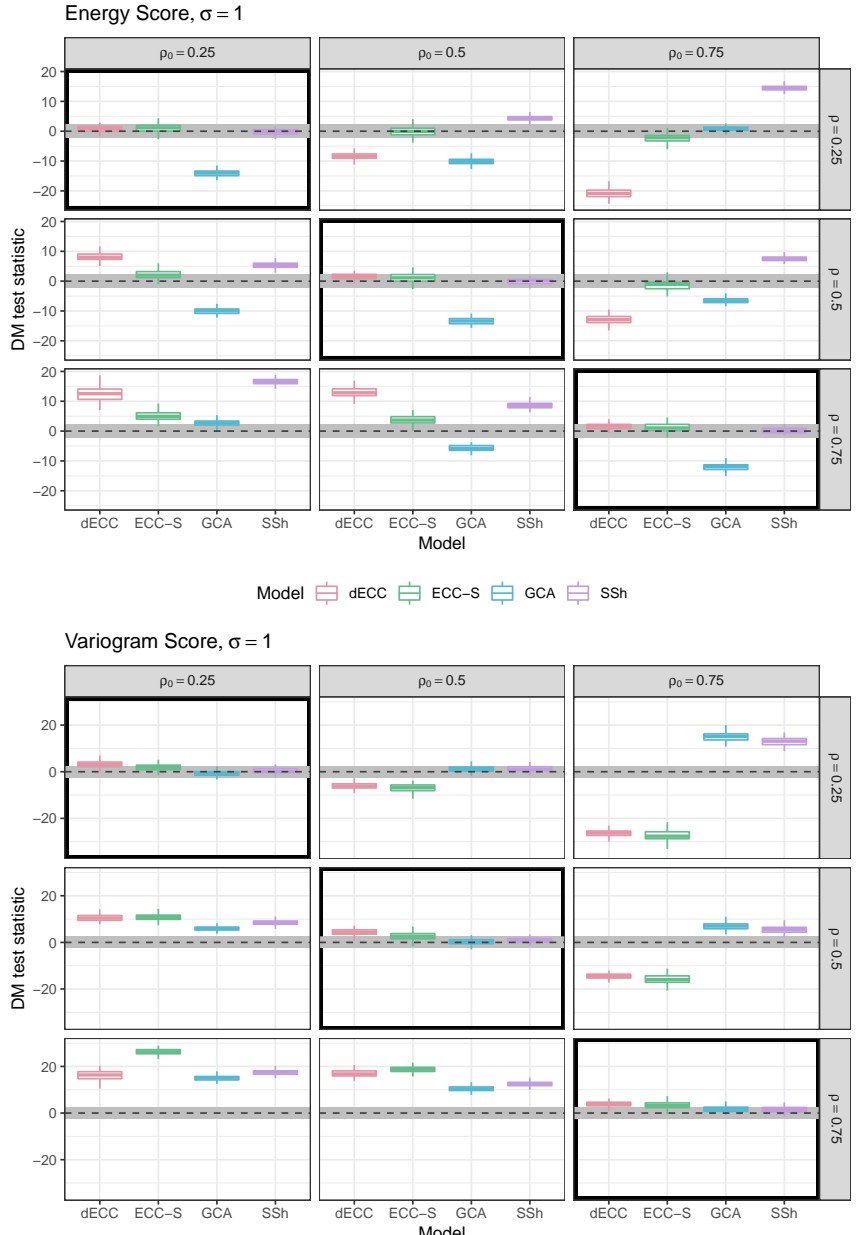

**Figure 5.** Summaries of DM test statistic values based on the ES (top) and the VS (bottom) for Setting 2 with $\epsilon = 3$, $\mu_0 = 2$, and $\sigma = 1$. ECC-Q forecasts are used as reference model such that positive values of the test statistic indicate improvements over ECC-Q and negative value indicated deterioration of forecast skill. Boxplots summarize results of the 100 Monte Carlo repetitions of each individual experiment. The horizontal gray stripe indicates the acceptance region of the two-sided DM test under the null hypothesis of equal predictive performance at a level of 0.05. Simulation parameter choices where the correlation structure of the raw ensemble is correctly specified ($\rho = \rho_0$) are surrounded by black boxes.





GCA shows much better performance and outperforms ECC-Q in almost all cases. Even for cases where $\rho = \rho_0$, no significant differences in terms of VS can be detected.

As before, the results for ECC-S and dECC strongly depend on the misspecification of the correlation structure. Further, the results now additionally vary with the variance misspecification parameter $\sigma$ also for ECC-S. In terms of the ES, ECC-S provides improvements over ECC-Q for $\rho > \rho_0$ and similar forecast quality for $\rho \leq \rho_0$. However, the forecast quality deteriorates for larger values of $\sigma$ and the ECC-S forecasts are significantly worse than those of ECC-Q, even for cases where $\rho = \rho_0$. In terms of the VS, ECC-S also provides significant improvements over ECC-Q for $\rho > \rho_0$ and even provides the best forecasts among all models, but shows significantly worse performance for $\rho < \rho_0$ across all values of $\sigma$.

The results for dECC are similar for ES and VS. For both scoring rules and $\rho < \rho_0$, the values of the DM test statistics move from positive (improvement over ECC-Q) to negative (deterioration compared to ECC-Q) values with increasing $\sigma$. For $\rho > \rho_0$, they instead change from negative to positive values. While improvements over ECC-Q can be observed for cases with a correctly specified correlation structure of the ensemble ($\rho = \rho_0$) and $\sigma = 1$, dECC performs worse than ECC-Q for those cases if $\sigma \neq 1$.

### 4.2.3 Setting 3: Multivariate censored GEV distributions

The four considered scenarios in Table 1 constitute different types of deviation of the ensemble from the observation properties.

- In Scenario B the location is correctly specified, but scale and shape are misspecified such that ensemble forecasts have both larger scale and shape, resulting in a heavier right tail and slightly higher point masses at zero. This scenario is taken as a reference among the four considered ones and shown in Figure 6. Additional figures with results for the remaining scenarios are provided in the Supplemental Material. Multivariate post-processing improves considerably upon the raw ensemble. ECC-Q is outperformed by SSh and GCA only when the absolute difference between $\rho_y$ and $\rho_x$ becomes larger. As before, this is likely caused by the use of past observations to determine the dependence template by GCA and SSh which proves beneficial in comparison to ECC-Q in cases of a highly incorrect correlation structure in the ensemble. For correctly specified correlations (panels on the main diagonal in Figure 6), the relative performance of the methods does not depend on the actual value of correlation.

- In Scenario A the observation location is shifted from 0 to a positive value for the ensemble, the observation scale is larger, and shape smaller than in the ensemble. Therefore, the ensemble forecasts come from a distribution with smaller spread than the observations, which is also centered away from zero and has lower point mass at 0. In comparison to Scenario B there are more outliers, especially for ECC-S. In case of correctly specified correlations, the performance of the methods also does not depend on the actual value of correlation as in Scenario B. Notably, EMOS-Q here performs mostly similar to the ensemble, while in the other 3 scenarios it typically performs worse than the ensemble if $\rho_x > \rho_y$.

- In Scenario C the observation location is larger, the scale smaller, and the shape larger than in the ensemble distribution. This results in an observation distribution with a much more heavy right tail and a much larger point mass at 0 compared to the ensemble distribution. Here, post-processing models frequently offers no or only slight improvements over the


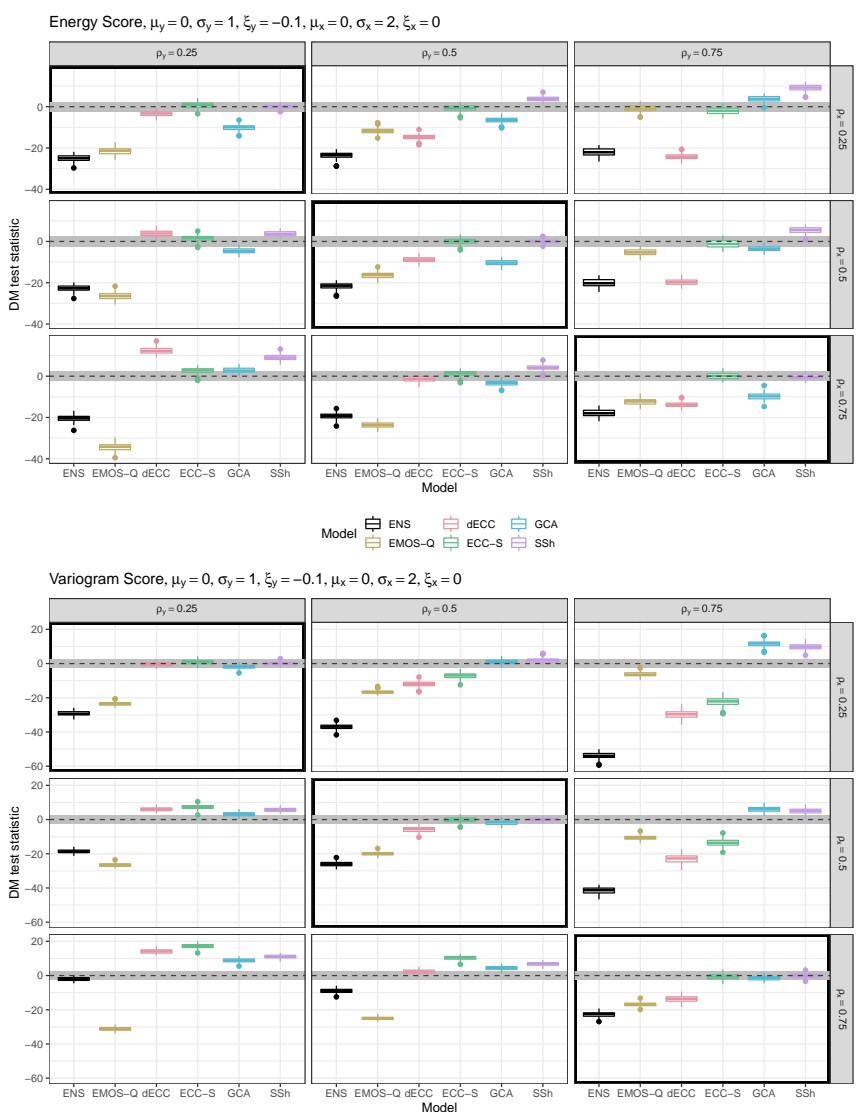

**Figure 6.** Summaries of DM test statistic values based on the ES (top) and the VS (bottom) for setting B from Table 1. ECC-Q forecasts are used as reference model such that positive values of the test statistic indicate improvements over ECC-Q and negative value indicated deterioration of forecast skill. Boxplots summarize results of the 100 Monte Carlo repetitions of each individual experiment. The horizontal gray stripe indicates the acceptance region of the two-sided DM test under the null hypothesis of equal predictive performance at a level of 0.05.





raw ensemble. While ECC-Q does not always outperform the raw ensemble forecasts, SSh still shows improved forecast

performance. As in the other scenarios, in case of correctly specified correlations, the performance of the methods does not depend on the actual value of correlation.

– In Scenario D all univariate distribution parameters are correctly specified. Therefore, the main differences in performance are imposed by the different misspecifications of the correlation structure. The main difference compared to the other scenarios is given by the markedly worse effects of not accounting for multivariate dependencies during post-

processing (EMOS-Q).

In general, the methods perform differently across the four scenarios, but for most situations multivariate post-processing improves upon univariate post-processing without accounting for dependencies. Furthermore, SSh reveals a good performance in all four scenarios when $\rho_y$ differs considerably from $\rho_x$. The performance of SSh has a tendency to improve further when the observation correlation is larger than the ensemble correlation. Within each of the four scenarios, the performance of

the methods is nearly identical in cases where the correlation is correctly specified. In other words, as long as the ensemble forecasts correctly represent the correlation of the observations, the actual value of the correlation does not have an impact on the performance of a multivariate post-processing method. Above described observations can be found both in terms of the ES, as well as the VS.

In addition to the scenarios from Table 1, further scenario variations were considered for $\rho_y = 0.75$ and $\rho_x = 0.25$, that

is for the case where ensemble correlation is too low compared to the observations. Figure 7 shows the situation where the observation location is larger, the scale smaller, but the shape also smaller than in the ensemble forecasts. This contrasts the situation in Scenario C. While in C the observations were heavier tailed with higher point mass at 0, here it is the other way around (the ensemble distribution is heavier tailed with higher point mass at 0). In accordance to Scenarios A, B, C (where there are parameter misspecifications in the ensemble compared to the observations), EMOS-Q performs better than the raw

ensemble and also better than dECC (as in B and C), while SSh and GCA perform best. However, by contrast to results in terms of the ES, GCA exhibits an even better performance compared to the other models in terms of the VS. This indicates that the VS might be better able to account for the correctly specified (or by post-processing improved) correlation structure than the ES.

### 4.2.4 Setting 4: Multivariate Gaussian distribution with changes over iterations

Figure 8 shows results in terms of ES and VS. We again only show results for $\rho, \rho_0 \in \{0.25, 0.5, 0.75\}$ and refer to the Supplementary Material for further results. The most notable differences compared to Setting 1 are that the different ECC variants here significantly outperform GCA and SSh not only for ensemble forecasts with correctly specified correlation structure, but also for small deviations of $\rho$ from $\rho_0$. Significant ES differences in favor of SSh are only obtained for large absolute differences of $\rho$ and $\rho_0$. Similar observations hold for GCA which, however, generally exhibits worse performance compared to SSh. The

ES differences among the ECC variants are only minor and usually not statistically significant.


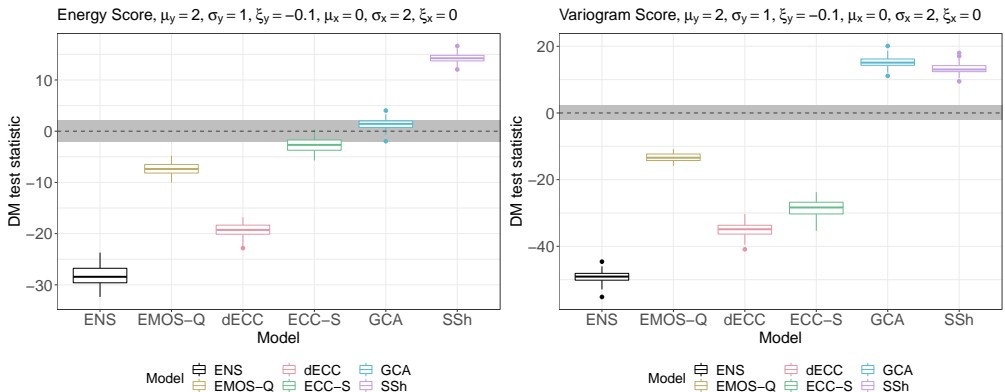

**Figure 7.** As Figure 6, but based on ES and VS for $(\mu_y, \xi_y, \sigma_y) = (2.0, -0.1, 1.0)$ and $(\mu_x, \xi_x, \sigma_x) = (0.0, 0.0, 2.0)$, where $\rho_y = 0.75$ and $\rho_x = 0.25$

Similar conclusions apply for the VS, however, GCA generally performs better than SSh, and ECC-S provides significantly worse forecasts compared to the other ECC variants for $\rho < \rho_0$.

Note that the main focus here was to demonstrate that in (potentially more realistic) settings with changes over iterations, naive implementations of the Schaake shuffle can perform worse than ECC variants. However, similarity-based implementa-
tions of the Schaake shuffle (Schefzik, 2016; Scheuerer et al., 2017) are available and may be able to alleviate this issue.

## 5   Discussion and conclusion

State of the art methods for multivariate ensemble post-processing were compared in four simulation settings which aimed to mimic different situations and challenges occurring in practical applications. Across all settings, the Schaake shuffle constitutes a powerful benchmark method that proves difficult to outperform, except for naive implementations in the presence of structural
change. By contrast to SSh, the Gaussian copula approach typically only provides improvements over variants of ensemble copula coupling if the parametric assumption of a Gaussian copula is satisfied or if forecast performance is evaluated with the variogram score. Results in terms of the CRPS further highlight an additional potential disadvantage in that the univariate forecast errors are larger compared to the competitors.

Not surprisingly, variants of ensemble copula coupling typically perform the better the more informative the ensemble
forecasts are about the true multivariate dependence structure. A particular advantage compared to standard implementations of SSh and GCA illustrated in Setting 4 may be given by the ability to account for flow-dependent differences in the multivariate dependence structure if those are present in the ensemble predictions, but not in a randomly selected subset of past observations.

There is no consistently best method across all simulation settings and potential misspecifications among the different ECC variants investigated here (ECC-Q, ECC-S and dECC). ECC-Q provides a reasonable benchmark model and will rarely yield
the worst forecasts among all ECC variants. Significant improvements over ECC-Q may be obtained by ECC-S and dECC,


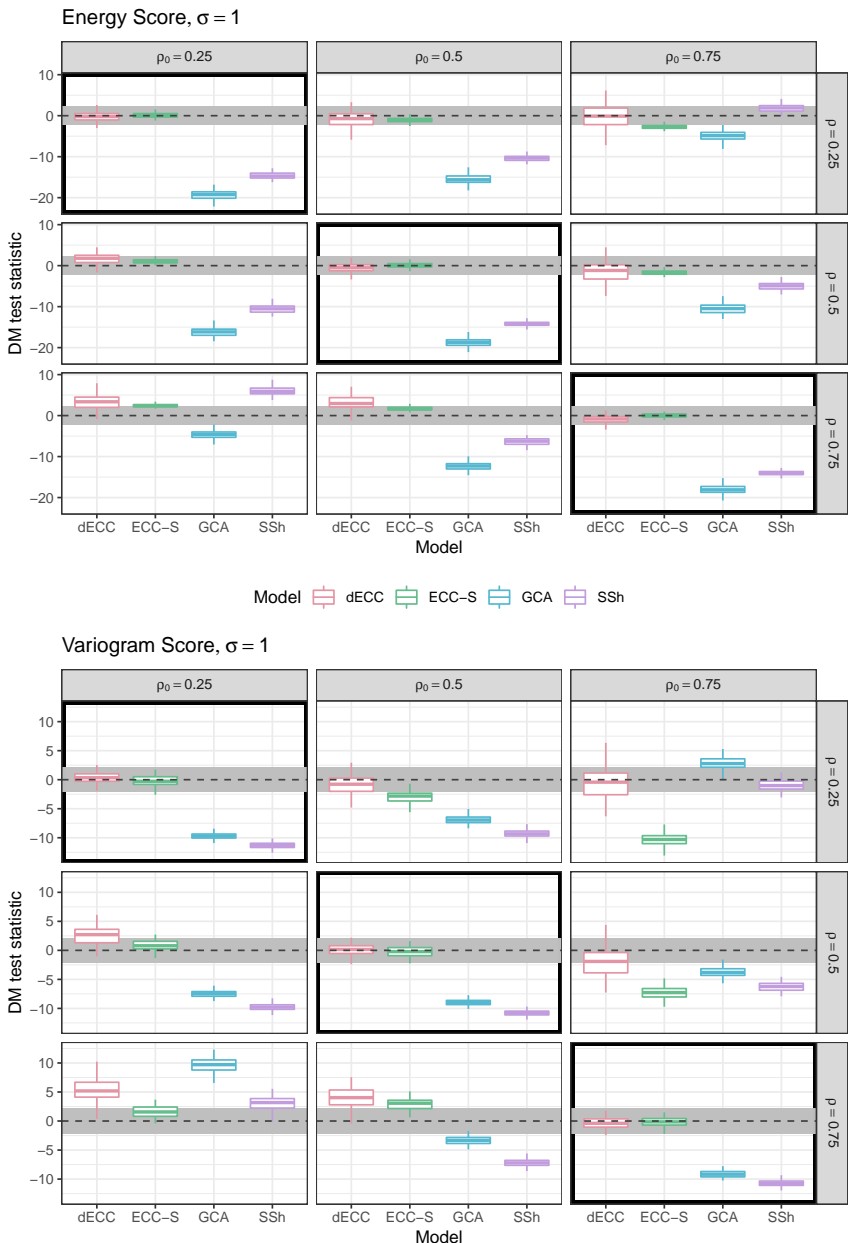

**Figure 8.** Summaries of DM test statistic values based on the ES (top) and the VS (bottom) for Setting 4 with $\epsilon = 1$ and $\sigma = 1$. ECC-Q forecasts are used as reference model such that positive values of the test statistic indicate improvements over ECC-Q and negative value indicated deterioration of forecast skill. Boxplots summarize results of the 100 Monte Carlo repetitions of each individual experiment. The horizontal gray stripe indicates the acceptance region of the two-sided DM test under the null hypothesis of equal predictive performance at a level of 0.05. Simulation parameter choices where the correlation structure of the raw ensemble is correctly specified ($\rho = \rho_0$) are surrounded by black boxes.





however, the results will strongly depend on the exact misspecification of the variance-covariance structure of the ensemble as well as the performance measure chosen for multivariate evaluation.

In light of the presented results it seems to be generally advisable to first test the Schaake shuffle along with ECC-Q. If structural assumptions on specific misspecifications of the ensemble predictions seem appropriate, extensions by other variants
of ECC or GCA might provide improvements. However, it should be noted that the results for real-world ensemble prediction systems may be influenced by many additional factors, and may differ when considering station-based or grid-based post-processing methods. The computational costs of all presented methods are not only negligible in comparison to the generation of the raw ensemble forecasts, but also compared to the univariate post-processing as no numerical optimization is required. It may thus be generally advisable to compare multiple multivariate post-processing methods for the specific dataset and
application at hand.

The simulation settings considered here provide several avenues for further generalization and analysis. For example, a detailed investigation of the effect of the dimension $d$ and a comparison of forecast quality in terms of multivariate calibration (Thorarinsdottir et al., 2016; Wilks, 2017) is left for future work. Further, the autoregressive structure of the correlations across dimensions may be extended towards more complex correlation functions, see, e.g., Thorarinsdottir et al. (2016, Section 4.2).
While we only considered multivariate methods based on a two step procedure combining univariate post-processing and dependence modeling via copulas, an extension of the comparison to parametric approaches along the lines of Feldmann et al. (2015) and Baran and Möller (2015) present another starting point for future work. Note that within the specific choices for Setting 1, the spatial EMOS approach of Feldmann et al. (2015) can be seen as a special case of GCA.

We have limited our investigation to simulation studies only as those settings allow to readily assess the effects of different
types of misspecifications of the various multivariate properties of ensemble forecasts and observations, and may thus help to guide implementations of multivariate post-processing. An important aspect for future work is to complement the comparison of multivariate post-processing methods by studies based on real-world datasets of ensemble forecasts and observations. However, the variety of application scenarios, methods and implementation choices likely requires large-scale efforts, ideally based on standardized benchmark datasets. A possible intermediate step might be given by the use of simulated datasets obtained
via stochastic weather generators (see, e.g., Wilks and Wilby, 1999) which may provide arbitrarily large datasets with possibly more realistic properties than the simple settings considered here.

A different perspective on the results presented here concerns the evaluation of multivariate probabilistic forecasts. In recent work Ziel and Berk (2019) argue that the use of Diebold-Mariano tests is of crucial importance for appropriately assessing the discrimination ability of multivariate proper scoring rules and find that the ES might not have as bad discrimination ability
as indicated by earlier research. The simulation settings and comparisons of multivariate post-processing methods considered here may be seen as additional simulation studies for assessing the discrimination ability of multivariate proper scoring rules. In particular, the results in Section 4 are in line with the findings of Ziel and Berk (2019) in that the ES does not exhibit inferior discrimination ability compared to the VS. Nonetheless, the ranking of the different multivariate post-processing methods strongly depends on the proper scoring rule used for evaluation, and further research on multivariate verification is required to
address open questions, improve mathematical understanding and guide model comparisons in applied work.





*Code availability.* `R` code with implementations of all simulation settings as well as code to reproduce the results presented here and in the Supplemental Material is available from https://github.com/slerch/multiv_pp.

## Appendix A:  Details on the left-censored generalized extreme value distribution

When the GEV distribution is left-censored at zero, its cumulative distribution function can be written as

$$F(y) = \begin{cases} e^{-t(y)}, & y \geq 0 \\ 0, & y < 0 \end{cases}, \quad \text{where} \quad t(y) = \begin{cases} \left(1 + \xi\left(\frac{y-\mu}{\sigma}\right)\right)^{-1/\xi}, & \xi \neq 0 \\ e^{-(y-\mu)/\sigma}, & \xi = 0 \end{cases}$$

for $y \in \mathfrak{Y}$, where $\mathfrak{Y} = [\mu - \sigma/\xi, \infty)$ when $\xi > 0$, $\mathfrak{Y} = (-\infty, \infty)$ when $\xi = 0$ and $\mathfrak{Y} = (-\infty, \mu - \sigma/\xi]$ when $\xi < 0$. This describes a three-parameter distribution family, where $\mu \in \mathbb{R}$, $\sigma > 0$, and $\xi \in \mathbb{R}$ are location, scale, and shape of the non-censored GEV distribution, respectively.

### Expectation and variance

Let $Y$ be a random variable distributed according to GEV and censored at zero to the left. From the law of total expectation

$$\mathrm{E}(g(Y)) = P(Y = 0)\mathrm{E}(g(Y)|Y = 0) + P(Y > 0)\mathrm{E}(g(Y)|Y > 0),$$

where the second term in the sum is given by

$$\mathrm{E}(g(Y)\mathbb{1}_{\{Y>0\}}) = \int_0^\infty g(y) f_Y(y)\,dy.$$

Here, $\mathbb{1}$ denotes the indicator function, $g$ is any function of $Y$ such that $g(Y)$ is a random variable, and $f_Y$ is the probability density function (PDF) of the non-censored GEV. By noting that $\mathrm{E}(Y|Y = 0) = \mathrm{E}(Y^2|Y = 0) = 0$, expectation and variance of the left-censored GEV can be computed from the two integrals $\int_0^\infty y f_Y(y)\,dy$ and $\int_0^\infty y^2 f_Y(y)\,dy$, the former existing when $\xi < 1$ and the latter existing when $\xi < 0.5$. Both integrals are not derived analytically here, but evaluated by numerical integration. In contrast to the non-censored GEV distribution, the variance of the left-censored version also depends on the parameter $\mu$, since different choices of $\mu$ lead to different left-censored CDFs which are not merely distinguished by location. Therefore $\mu$ is a location parameter for the non-censored GEV, but not for the left-censored version.

## Appendix B:  Evaluating probabilistic forecasts

### B1   Proper scoring rules

The comparative evaluation of probabilistic forecasts is usually based on proper scoring rules. A proper scoring rule is a function

$\mathrm{S} : \mathcal{F} \to \mathbb{R},$





which assigns a numerical score $S(F,y)$ to a pair of a forecast distribution $F \in \mathcal{F}$ and a realizing observation $y \in \Omega$. Here, $\mathcal{F}$ denotes a class of probability distributions supported on $\Omega$. The forecast distribution $F$ may come in the form of a predictive CDF, PDF, or a discrete sample as in the case of ensemble predictions. A scoring rule is called proper if

$$\mathbb{E}_G \mathrm{S}(G,Y) \leq \mathbb{E}_G \mathrm{S}(F,Y)$$

for all $F, G \in \mathcal{F}$, and strictly proper if equality holds only if $F = G$. See Gneiting and Raftery (2007) for a review of proper
scoring rules from a statistical perspective.

The most popular example of a univariate (i.e., $\Omega \subset \mathbb{R}$) proper scoring rule in the environmental sciences is given by the continuous ranked probability score (CRPS),

$$\mathrm{CRPS}(F,y) = \int_\Omega \left(F(z) - \mathbb{1}\{z \geq y\}\right)^2 \mathrm{d}z.$$

Over the past years a growing interest in multivariate proper scoring rules accompanies the proliferation of multivariate
probabilistic forecasting methods in applications across disciplines. The definition of proper scoring rules from above straightforwardly extends towards multivariate settings (i.e., $\Omega \subset \mathbb{R}^d$). A variety of multivariate proper scoring rules has been proposed over the past years, usually focused on cases where multivariate probabilistic forecasts are given as samples from the forecast distributions.

To introduce multivariate scoring rules let $\mathbf{y} = (y^{(1)}, \ldots, y^{(d)}) \in \Omega \subset \mathbb{R}^d$, and let $F$ denote a forecast distribution on $\mathbb{R}^d$
given through $m$ discrete samples $\mathbf{X}_1, \ldots, \mathbf{X}_m$ from $F$ with $\mathbf{X}_i = (X_i^{(1)}, \ldots, X_i^{(d)}) \in \mathbb{R}^d, i = 1, \ldots, m$. Important examples of multivariate proper scoring rules include the energy score (ES; Gneiting et al., 2008),

$$\mathrm{ES}(F,y) = \frac{1}{m}\sum_{i=1}^m \|\mathbf{X}_i - \mathbf{y}\| - \frac{1}{2m^2}\sum_{i=1}^m\sum_{j=1}^m \|\mathbf{X}_i - \mathbf{X}_j\|,$$

where $\|\cdot\|$ is the Euclidean norm on $\mathbb{R}^d$, and the variogram score of order $p$ ($\mathrm{VS}^p$; Scheuerer and Hamill, 2015),

$$\mathrm{VS}^p(F,y) = \sum_{i=1}^d\sum_{j=1}^d w_{i,j}\left(\left|y^{(i)} - y^{(j)}\right|^p - \frac{1}{m}\sum_{k=1}^m \left|X_k^{(i)} - X_k^{(j)}\right|^p\right)^2.$$

Here, $w_{i,j}$ is a non-negative weight that allows to emphasize or down-weight pairs of component combinations, and $p$ is the order of the variogram score. Following suggestions of Scheuerer and Hamill (2015), we considered $p = 0.5$ and $p = 1$. As none of the simulations settings indicated any substantial differences, we set $p = 1$ throughout and denote $\mathrm{VS}^1(F,y)$ by $\mathrm{VS}(F,y)$. Since the generic multivariate structure of the simulation settings does not impose any meaningful structure in pairs of components we focus on the unweighted versions of the variogram score. Several weighting schemes have been tested, but
did not lead to any substantially different conclusions.

We utilize implementations provided in the R package `scoringRules` (Jordan et al., 2019) to compute univariate and multivariate scoring rules for forecast evaluation and post-processing model estimation.





## B2   Diebold-Mariano tests

Statistical tests of equal predictive performance are frequently used to assess the statistical significance of observed score
differences between models. We focus on Diebold-Mariano (DM; Diebold and Mariano, 1995) tests which are widely used in
the econometric literature due to their ability to account for temporal dependencies. For applications in the context of post-
processing, see, e.g., Baran and Lerch (2016).

For a (univariate or multivariate) proper scoring rule S and sets of two competing probabilistic forecasts $F_i$ and $G_i$, $i = 1, \ldots, n_{\text{test}}$ over a test set, the test statistic of the DM test is given by

$$T_{n_{\text{test}}}^{\text{DM}} = \sqrt{n_{\text{test}}} \frac{\overline{S(F, y)} - \overline{S(G, y)}}{\hat{\sigma}}, \tag{B1}$$

where $\overline{S(F, y)} = \frac{1}{n_{\text{test}}} \sum_{i=1}^{n_{\text{test}}} S(F_i, y_i)$ and $\overline{S(G, y)} = \frac{1}{n_{\text{test}}} \sum_{i=1}^{n_{\text{test}}} S(G_i, y_i)$ denote the mean score values of $F$ and $G$ over the
test set of size $n_{\text{test}}$, respectively. In (B1), $\hat{\sigma}$ denotes an estimator of the asymptotic standard deviation of the sequence of score
differences of $F$ and $G$. Positive values of $T_{n_{\text{test}}}^{\text{DM}}$ indicate a superior performance of $G$, whereas negative values indicated a
superior performance of $F$.

Under standard regularity assumptions and the null hypothesis of equal predictive performance, $T_{n_{\text{test}}}^{\text{DM}}$ asymptotically follows
a standard normal distribution which allows to assess the statistical significance of differences in predictive performance. We
utilize implementations of DM tests provided in the R package `forecast` (Hyndman and Khandakar, 2008).

*Author contributions.*   All authors jointly discussed and devised the design and setup of the simulation studies. A variant of Setting 1 was first
investigated in a MSc thesis written by MG (Graeter, 2016), co-supervised by SL. SL wrote evaluation and plotting routines, implemented
simulation settings 1 and 4, partially based on code and suggestions from MG and SH, and provided a simulation framework in which Setting
2 (SB) and Setting 3 (AM and JG) were implemented. All authors jointly analyzed the results and edited the manuscript, coordinated by SL.

*Competing interests.*   Sebastian Lerch and Stephan Hemri are editors of the special issue on "Advances in post-processing and blending of
deterministic and ensemble forecasts". The remaining authors declare that they have no conflict of interest.

*Acknowledgements.*   The authors gratefully acknowledge support by the Deutsche Forschungsgemeinschaft (DFG) through project MO-
3394/1-1 "Statistische Nachbearbeitung von Ensemble-Vorhersagen für verschiedene Wettervariablen". Sebastian Lerch is further supported
by DFG through SFB/TRR 165 "Waves to Weather", and Sándor Baran by the National Research, Development and Innovation Office under
Grant No. NN125679. The authors thank Tilmann Gneiting and Kira Feldmann for helpful discussions.



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
