# Peer review of "Simulation-based comparison of multivariate ensemble post-processing methods"

_Nonlinear Processes in Geophysics, 2019_

## Referee Comment (RC1) · Anonymous Referee #1 · 25 Jan 2020

This paper reports the results of a comprehensive simulation study comparing different methods for modeling spatial, temporal, and inter-variable correlations in statistically postprocessed ensemble forecasts. With a number of new multivariate methods having been developed in recent years, a study like this is of great interest as it allows readers to get an overview of the strengths and limitations of the different approaches.

General comments:

1. With the goal of this paper being a comparison between different methods for multivariate ensemble postprocessing, I feel that a more detailed discussion of the key features (e.g. optimal sampling of the predictive distribution vs. random sampling, as-

sumption of stationarity of the copula structure vs. flow dependent copula structure, etc.) of the different methods should be given (possibly in the form of a table). This could serve as a motivation for the different simulation settings, which try to mimic situtations where some of the assumptions are met while others are not.

2. Related to 1., I feel that the role of ensemble size (which has a big impact on the representation of the multivariate distribution) should be discussed a bit more. This seems relevant as for some methods it is easy to generate an ensemble of any size while for others it is not. I'm not suggesting that additional experiments should be performed, but a brief discussion of the findings in Wilks (2015) could be useful in a context where strengths and limitations of different multivariate postprocessing approaches are compared.

3. Why has so much focus been given to simulation settings that are based on a time-independent model for the simulated forecasts and observations? I would argue that this (time-independence) is not a feature commonly encountered in applications, but with 3 out of 4 settings being time-independent, multivariate methods that assume a stationary copula structure could be perceived as being more versatile than they really are. Agreed that setting 1 is a natural starting point for such a comparison and that setting 3 is interesting because of the entirely different nature of the marginal distributions (skewness, possibility of heavy tails, mixed discrete-continuous distributions), but what do we learn from setting 2 that we cannot learn from 1 and 3? The main difference to 1 seens to be the poorer performance of GCA, but an explanation for this is not given, and so the insights gained from this setting are limited.

Specific comments:

131-132: Please check if that statement is correct. In my recollection the selection of past observations in Clark et al. (2004) was not random, but was based on the valid date of the forecast

293-297: I find setting 4 the most interesting, but I find the particular definitions of the

model parameters unnecessarily complicated. Specifically, I don't get a good intuition of what kind of time-varying correlations this model implies. Couldn't one simply define:

$\Sigma_{i,j} = \sigma \rho^{|i-j\}}$

as in the other settings, but now make $\rho$ time-varying, e.g. via

$\rho(t) = \rho_0(1 - a/2) + \rho_0(a/2)\sin(\frac{2\pi t}{n}), a \in (0,1)$

This model would be autoregressive with lag-1 correlations oscillating between $\rho_0$ and $\rho_0(1 - a)$, and thus have a more intuitive interpretation.

260-264: I find this notation a bit confusing since previously the subscript/superscript 'O' was used for observations and here the subscript '0' (which is hard to discern from 'O' in the NPG font) is used to denote the fraction of zero values. The notation is also inconsistent in that in setting 3 'x' and 'y' are used to denote forecasts and observations, in contrast to the subscript/superscript 'O' for observations in the other settings.

323 '... are identical to those of ECC-Q ..': Is this really true for ECC-S? The way it is described here, ECC-S seems to imply some level of randomization (albeit less than ECC-R), so the sampling is not the same as for ECC-Q.

Fig. 2: I don't think that this figure is really necessary. Why is the (univariate) performace of ECC-Q compared to the raw ensemble relevant to the comparison of multivariate postprocessing approaches?

354 '... SSh never performs substantially worse ...': Why would we expect otherwise? The only drawback of SSh in the present context is the underlying assumption of time-invariance of the correlation structure, which is not a drawback in a time-invariant simulation setting. If not discussed before, this paragraph could be an opportunity to discuss this issue of time-invariant vs time-varying correlations.

360: I wonder if ECC-S gives the better results for the wrong reason here. Maybe I am misunderstanding its key idea, but to me this is essentially a compromise between

ECC-Q and ECC-R. Is it possible that the small amount of randomization in ECC-S weakens the correlations, which is beneficial for $\rho > \rho_0$ and detrimental for $\rho < \rho_0$ ? An argument against this hypothesis is that the performance of ECC-S is not significantly different from ECC-Q when $\rho = \rho_0$. I just cannot think of any good reason why ECC-S would be better than ECC-Q. If the authors have any explanation for these results I would encourage them to include those in the discussion of the results.

Fig. 6: The caption should state that this is scenario B from setting 3

458 'changes over iterations': I would change this terminology and speak of 'time' instead of 'iterations', and be more specific about what changes/varies over time. Likewise in 464-465 I would clarify what you mean by 'structural change'

Language and typos:

43: ... studies allow one to specifically tailor ...

135: sufficiently many

137: 'not directly straightforward' sounds weird, I'd just say 'not straightforward'

249: ... allows one to generate ...

375: 'can potentially be explained' sounds weird, maybe better 'may be explained'

388: I think you want to say 'In terms of ...'

443: Change to 'the other way round' and 'In accordance with'

445: Change to 'in contrast to'

References:

D.S. Wilks (2015): Multivariate ensemble Model Output Statistics using empirical copulas. QJRMS 141(688), 945-952.

[Figure]

2019-62, 2020.

---

## Referee Comment (RC2) · Anonymous Referee #2 · 6 Feb 2020

In this paper, a comprehensive simulation study is implemented, comparing multiple multivariate statistical post-processing methods which all combine standard univariate post-processing with one of four techniques to reintroduce spatial, temporal or inter-variable dependencies. It is well-written, an important contribution given the variety of techniques available and potentially very useful to identify optimal operational post-processing strategies for varying types of data. Just a few clarifications and changes are needed.

[Figure]

**General comments**

- I would have liked to have more focus (or at least comments) on the effect of ensemble size and dimension. The here chosen ensemble of 50 members is at the upper limit of what is now operationally produced, with the majority far below this number. Of course it is important for the study to have a sufficient number of data points so as to produce significant results, but it would also be interesting to look at settings with a smaller number of ensemble members. Also, the number of dimensions ranges from 4 to 5, which would correspond to looking at consistency between a few weather variables, but would usually be too low for a setting where preserving spatial or temporal features are important. I wonder if the findings would be different for smaller ensembles or higher dimensions.

- I find it very interesting that the performance of certain methods is sometimes very different when $\rho > \rho_0$ than in the opposite case. Do you have any explanation for this?

**Specific comments**

1. Line 71 and Line 153: The correlation matrix here is not necessarily the identity matrix, so I don't think it is a standard normal distribution.

2. Line 74: I would mention here that $m$ is the number of ensemble members.

3. Section 2.2: There is a mixture of $x$ and $X$ used to define samples and ensemble forecasts, but I was confused why this distinction is made within the notation, it seems inconsistent.

4. Lines 184-186: For the other settings it is mentioned to which weather variables these settings could apply. It would be nice to add something like this to Setting 1, as well.

5. Line 242: The notation in this setting is different from the others and this is confusing. Here, the forecasts and observations are marked with x and y, whereas the other settings use o/0 to mark the observations.

6. Line 276: Is there a specific reason, why $d$ is 4 in this setting and 5 in the others?

7. Lines 292-293: Some of the matrices are in bold face, some are not.

8. Figure 1: I am a bit surprised to see that GCA is performing that much worse as compared to the other post-processing methods (in a univariate sense). There are even cases where the performance is equal or possibly worse than for the raw ensemble. Do you have an idea why that could be?

9. Lines 328-330: Can you explain a bit further what you mean by "optimal in the terms of the CRPS"?

10. Lines 334-335: Naturally, scenario D has the smallest improvement compared to the others. Does that also mean that the scenarios are on the same absolute skill level after post-processing?

11. Footnote 4: In my opinion it would be clearer if you refer to ECC-Q as EMOS-Q in this section as well.

12. Lines 415-430: Can you refer to the figures in the appendix that show these results by number?

13. Line 446: "the VS might be better able to account..." This is confirming a known result, therefore "might" is a bit unsuitable.

**Technical corrections**

1. Line 327: I would move the sentence beginning "Note that" to footnote 4, as it directly relates to the changes in the marginal distributions mentioned there.

2. Line 356: I find this sentence a bit confusing. Should there be a comma before "the less information"?

3. Line 386: Missing comma before "where".

4. Lines 415 and 441: I would add "parameter" after "observation location".

---

## Referee Comment (RC3) · Zied Ben Bouallegue (Referee) · 7 Feb 2020

This paper present results of an intercomparison study. Different multivariate ensemble post-processing methods are compared using toy-model simulations. The focus is on empirical copula methods that are generally applied as a second post-processing step, after univariate post-processing step, in order to provide coherent multivariate structure to ensemble calibrated forecasts.

As the reviewer is the developer of one of the methods compared here, he prefers to make himself know. There exists no conflict of interest (stricto sensu) but potential cognitive biases from the reviewer side when scrutinizing the results.

The paper is clear, well structured, and well written. However, the choice regarding the selection of illustrations is not sufficiently motivated to my opinion. This choice is important because it drives the discussion and the main conclusion of the study. A 3-point argument is developed below to explain this criticism.

You claim in the conclusion (L462/463) that the 4 simulation settings aim to mimic different situations and challenges occurring in practical situations. However, the link with practical situations is sometimes weak or missing. In particular, it would be interesting to link misspecification definitions with practical examples. What sigma > 1 and sigma < 1 mean, and similarly what does rho > rho0 and rho < rho0 mean and "look like" in practice? In which situations should one expect to encounter these types of misspecification? Which type of misspecification situations are the most common in practice? Are combinations of misspecified elements more common than others (sigma <0 and rho< rho0 for example)?

Once the link to the applications is clarified, illustrations could be chosen consistently. Result material is abundant, so one selection criterion could be to focus on the main misspecification encountered in practical situations. For example, you use sigma equals 1 to illustrate results in Setting 2. Does that often occur in practice? One could rather illustrate Setting 2 with sigma<1. Similarly, for Setting 3, scenario B is used for illustration purposes. It corresponds to the case where the ensemble forecasts have a heavier right tails and slightly higher point mass at zero than the observations. Does that often occur in practice? No justification for the use of this scenario as reference is provided. Scenario A (the ensemble comes from a distribution with smaller spread) or Scenario C (the observation distribution has a much heavier right tail) seem to be more likely to be faced in practice.

Your conclusion points to the robustness of the SSh method and so you encourage post-processing practitioners to consider this method as their first choice. Based on the results presented in the manuscript and the ones in the supplemental material document, one could draw the opposite recommendation. First choice methods are

available for different types of misspecification and so one could encourage to apply SSh only when the misspecification is unknown or difficult to identify. Specific recommendations would read:

1. If the ensemble is underdispersive and the ensemble correlation is too weak, d-ECC is the best option, both in terms of ES and VS.

2. If the ensemble is underdispersive and the ensemble correlation is too strong, ECC-S is the best option in terms of VS and one of the best options in terms of ES.

3. If the ensemble is overdispersive and the ensemble correlation is too strong, d-ECC is the best option, both in terms of ES and VS.

4. If the ensemble is overdispersive and the ensemble correlation is too weak, GCA is the best option in terms of VS, but is less performant in terms of ES. GCA proves to work well even when the overdispersiveness error characteristic is relaxed.

5. Otherwise consider using SSh, in particular when the correlation structures in the forecasts and observations are very dissimilar.

This is valid for all types of distributions (so all types of weather variables). Are these conclusions still valid in case of time varying misspecification? Verification results are missing to conclude here. Therefore, the authors are encouraged to investigate various sigmas in Setting 4 in order to collect evidences, and could draw conclusions accordingly.

---

## Author Comment (AC1) · 2 May 2020

*We thank the three reviewers for their positive assessment and their thoughtful comments which, we believe, will strengthen the manuscript. Below we addressed each comment in turn. Our replies are in italics. We also created a track-changes PDF for your convenience.*

*Kind regards,*
*Sebastian Lerch, Sándor Baran, Annette Möller, Jürgen Groß, Roman Schefzik, Stephan Hemri and Maximiliane Graeter*

**Reviewer 1**

This paper reports the results of a comprehensive simulation study comparing different methods for modeling spatial, temporal, and inter-variable correlations in statistically postprocessed ensemble forecasts. With a number of new multivariate methods having been developed in recent years, a study like this is of great interest as it allows readers to get an overview of the strengths and limitations of the different approaches.

General comments:

1. With the goal of this paper being a comparison between different methods for multivariate ensemble postprocessing, I feel that a more detailed discussion of the key features (e.g. optimal sampling of the predictive distribution vs. random sampling, assumption of stationarity of the copula structure vs. flow dependent copula structure, etc.) of the different methods should be given (possibly in the form of a table). This could serve as a motivation for the different simulation settings, which try to mimic situations where some of the assumptions are met while others are not.

*Thank you for this suggestion. We have added a table at the beginning of Section 2.2 where we now compare several key features of the different multivariate post-processing methods. We further refer to the related findings in Wilks (2015) (next comment) at the beginning of Section 2.2 and in the Discussion.*

2. Related to 1., I feel that the role of ensemble size (which has a big impact on the representation of the multivariate distribution) should be discussed a bit more. This seems relevant as for some methods it is easy to generate an ensemble of any size while for others it is not. I'm not suggesting that additional experiments should be performed, but a brief discussion of the findings in Wilks (2015) could be useful in a context

where strengths and limitations of different multivariate postprocessing approaches are compared.

*We agree that the role of ensemble size is important and warranted a more extensive discussion. We have performed additional simulations for Settings 1 and 3 (Settings 1 and 2 in the revised paper) with ensemble sizes between 5 and 100. For Setting 1, the results are largely as it can be expected: The relative differences in terms of the ES between ECC-Q and ECC-S, and between ECC-Q and GCA become increasingly negligible with increasing ensemble size; likely due to increasingly smaller intervals from which random quantiles are drawn. Further, SSh shows improved predictive performance for larger numbers of ensemble members for $\rho_0 < \rho$ in case of the ES, and for for $\rho_0 > \rho$ in case of the VS. This can likely be attributed to the corresponding increase in sample sizes when determining the dependence templates, similar to the effects on GCA. The relative performance of dECC is strongly effected by changes in $m$ for large misspecifications in the correlation parameters. A positive effect of larger numbers of members relative to ECC-Q in terms of both scoring rules can be detected for $\rho_0 > \rho$ when $\sigma < 1$, and for $\rho_0 < \rho$ when $\sigma > 1$. In both cases, the corresponding effects are negative, when the misspecification in $\sigma$ is reversed.*

*In Setting 2 (old Setting 3), in contrast to Setting 1, GCA seems to benefit most from an increasing number of members, while SSh benefits only slightly in terms of the ES, and stronger in terms of the VS. This is for example illustrated in the attached Figure 1, which corresponds to the additional scenario presented in Figure 5 in the main paper. As in Setting 1, ECC-S becomes more similar in terms of the ES to the reference ECC-Q for increasing number of members. However, this effect, is not (or not that strongly) observable for the VS, where the number of members has nearly no effect on ECC-S.*

*We have added a paragraph to Sections 4.2.1 and 4.2.2 where the effect of ensemble size is discussed shortly. Figures with additional results have been added to the Supplemental Material in Sections 1.1 and 2.2 there. We are aware that displaying*

*boxplots for several choices of $m$ within a single figure is not optimal since the underlying random samples within each panel will necessarily differ across different values of $m$. Nonetheless, we believe that the differences due to random sampling are negligible due to the application of DM tests and the consideration of 100 repetitions of each simulation experiment. Therefore we chose the illustration added to the Supplemental Material in Sections 1.1 and 2.2 because the effects of changes in $m$ are straightforward to compare.*

3. Why has so much focus been given to simulation settings that are based on a time-independent model for the simulated forecasts and observations? I would argue that this (time-independence) is not a feature commonly encountered in applications, but with 3 out of 4 settings being time-independent, multivariate methods that assume a stationary copula structure could be perceived as being more versatile than they really are. Agreed that setting 1 is a natural starting point for such a comparison and that set-ting 3 is interesting because of the entirely different nature of the marginal distributions(skewness, possibility of heavy tails, mixed discrete-continuous distributions), but what do we learn from setting 2 that we cannot learn from 1 and 3? The main difference to 1 seems to be the poorer performance of GCA, but an explanation for this is not given,and so the insights gained from this setting are limited.

*We agree that the results of Setting 2 were overall very similar to those of Setting 1. Therefore, we have removed Setting 2 from the paper and added a slightly adjusted version to the Supplemental Material. The former Setting 2 is there denoted by Setting S1, and setup and results are now discussed in Section 5 of the Supplemental Material. Accordingly, the paper has been adjusted as follows:*

- *Setting 4 is now Setting 3, and Setting 3 is now Setting 2*

- *Sections 3.2 and 4.2.2 have been removed from the paper and added to the Supplemental Material*

- *Figure 1 has been adjusted by removing results for Setting 2*

- *several references to Setting 2 in the text have been removed throughout*

- *A short paragraph referring to the additional setting (now in the Supplemental Material) has been added to the end of Section 3.1.*

Specific comments:

131-132: Please check if that statement is correct. In my recollection the selection of past observations in Clark et al. (2004) was not random, but was based on the valid date of the forecast

*Thank you for pointing this out, you are of course correct. The description of SSh has been corrected and information about the random selection of training cases has been added to Section 3.1 in step (S3).*

293-297: I find setting 4 the most interesting, but I find the particular definitions of the model parameters unnecessarily complicated. Specifically, I don't get a good intuition of what kind of time-varying correlations this model implies. Couldn't one simply define

$$\Sigma_{i,j} = \sigma \rho^{|i-j|}$$

as in the other settings, but now make $\rho$ time-varying, e.g. via

$$\rho(t) = \rho_0(1 - a/2) + \rho_0(a/2)\sin\left(\frac{2\pi t}{n}\right).$$

This model would be autoregressive with lag-1 correlations oscillating between $\rho_0$ and $\rho_0(1 - a)$, and thus have a more intuitive interpretation.

*Thank you for the suggestion of this alternative setting. We agree that this definition has a more intuitive interpretation. However, we believe that the way in which $\rho$ and $\rho_0$ are defined does not directly correspond to the situation we aimed to cover in Setting 4: Our goal was to mimic a situation in which the covariance structure of both observations and ensemble varies over time, with possible misspecifications of this structure in the ensemble. In the setup suggested above, the covariance structure of the observations does not change over time. Therefore, non-time-varying methods such as GCA and SSh which assume stationarity of the covariance structure are at an advantage in that they do not suffer from the (time-varying) misspecifications in the raw ensemble. An exemplary illustration is given in the attached Figure 2. Note that the rows show results for different values of $a$. SSh here never performs worse that ECC-Q and all methods significantly outperform ECC-Q in terms of the VS for almost all parameter values.*

*In order to provide a time-varying simulation with a possibly more intuitive interpretation, we have added a variant of the setting suggested above to the paper. In this new Setting 3B, we set $\Sigma_{i,j}^0(t) = \rho_0^{|i-j|}(t)$, for $i, j = 1, \ldots, d$, where the correlation parameter $\rho_0(t)$ varies over iterations according to*

$$\rho_0(t) = \rho_0 \cdot \left(1 - \frac{a}{2}\right) + \rho_0 \cdot \left(\frac{a}{2}\right) \sin\left(\frac{2\pi t}{n}\right)$$

*for $a \in (0, 1)$ for the observations, and similarly, $\Sigma_{i,j}(t) = \sigma \rho_{|i-j|}(t)$, for $i, j = 1, \ldots, d$, where*

$$\rho(t) = \rho \cdot \left(1 - \frac{a}{2}\right) + \rho \cdot \left(\frac{a}{2}\right) \sin\left(\frac{2\pi t}{n}\right)$$

*for the ensemble predictions. Correlations in both cases are autoregressive with lag-1 and oscillated between $\rho_0$ and $\rho_0(1 - a)$, and $\rho$ and $\rho(1 - a)$, respectively.*

*The description of Setting 3B was added to Section 3.3, results are discussed in Section 4.2.3, and a corresponding figure has been added to that section. Results for*

NPGD

*additional simulation parameters for Setting 3B are provided in the Supplemental Material.*

260-264: I find this notation a bit confusing since previously the subscript/superscript 'O' was used for observations and here the subscript '0' (which is hard to discern from 'O' in the NPG font) is used to denote the fraction of zero values. The notation is also inconsistent in that in setting 3 'x' and 'y' are used to denote forecasts and observations,in contrast to the subscript/superscript 'O' for observations in the other settings.

*The subscript '0' (to denote zero values) has been changed to subscript 'z'. The subscripts 'x' and 'y' have been changed to match the subscript notation in the other sections.*

323 '... are identical to those of ECC-Q ...': Is this really true for ECC-S? The way it is described here, ECC-S seems to imply some level of randomization (albeit less than ECC-R), so the sampling is not the same as for ECC-Q.

*Thank you for spotting this. The univariate distributions of ECC-S will indeed not be identical to those of ECC-Q. Comparing univariate results, we however found that the differences in terms of predictive performance are only very minor and were not noticeable in plots. We have modified the corresponding paragraph which now reads " Note that for ECC-S and SSh differences in the univariate forecast distributions compared to those of ECC-Q may arise from randomly sampling the quantile levels in ECC-S and due to random fluctuations due to the 10 random repetitions that were performed to account for simulation uncertainty of those methods. However, we found the effects on the univariate results to be negligible and omit ECC-S, dECC and SSh from Figure 1."*

Fig. 2: I don't think that this figure is really necessary. Why is the (univariate) perfor-

Interactive
comment
mance of ECC-Q compared to the raw ensemble relevant to the comparison of multivariate postprocessing approaches?

*Figure 2 and the corresponding text paragraph have been removed.*

354 '... SSh never performs substantially worse ...': Why would we expect otherwise?The only drawback of SSh in the present context is the underlying assumption of time-invariance of the correlation structure, which is not a drawback in a time-invariant simulation setting. If not discussed before, this paragraph could be an opportunity to discuss this issue of time-invariant vs time-varying correlations.

*We have added a short discussion to the corresponding paragraph.*

360: I wonder if ECC-S gives the better results for the wrong reason here. Maybe I am misunderstanding its key idea, but to me this is essentially a compromise between ECC-Q and ECC-R. Is it possible that the small amount of randomization in ECC-S weakens the correlations, which is beneficial for $\rho > \rho_0$ and detrimental for $\rho < \rho_0$? An argument against this hypothesis is that the performance of ECC-S is not significantly different from ECC-Q when $\rho = \rho_0$. I just cannot think of any good reason why ECC-S would be better than ECC-Q. If the authors have any explanation for these results I would encourage them to include those in the discussion of the results.

*Unfortunately, we are unable to provide a comprehensive explanation for these observations. As will be argued in the response to Reviewer 2 below, drawing conclusions on specific aspects of the multivariate methods or on the effect of single simulation parameters is difficult and is further impeded by a lack of well-understood multivariate evaluation methods. One potential advantage of the sampling scheme in ECC-S in comparison to the use of quantiles at fixed levels can become apparent when considering corresponding EMOS-S and EMOS-Q variants. When evaluated in terms of the VS, the random sampling in ECC-S may alleviate issues arising from over-estimated*

*correlations due to the use of the same quantile levels in EMOS-Q. However, it appears to not be straightforward how these observations will be effected by applying ECC, and what the effects of the 10 repetitions of each individual simulation experiments that were performed to account for simulation uncertainty, would be.*

Fig. 6: The caption should state that this is scenario B from setting 3

*Fixed.*

458 'changes over iterations': I would change this terminology and speak of 'time' instead of 'iterations', and be more specific about what changes/varies over time. Likewise in 464-465 I would clarify what you mean by 'structural change'

*Changed as suggested.*

Language and typos:

43: ... studies allow one to specifically tailor ...

*Fixed.*

135: sufficiently many

*Fixed.*

137: 'not directly straight forward' sounds weird, I'd just say 'not straightforward'

*Changed as suggested.*

249: ... allows one to generate ...

*Fixed.*

375: 'can potentially be explained' sounds weird, maybe better 'may be explained'

*Changed as suggested.*

388: I think you want to say 'In terms of ...'

*Fixed.*

443: Change to 'the other way round' and 'In accordance with'

*Changed as suggested.*

445: Change to 'in contrast to'

*Changed as suggested.*
* * *
[Figure]

**Fig. 1.** Effect of ensemble size in the GEV0 setting shown as grouped boxplots of ensemble sizes $m=5,20,50,100$ (5 corresponds to lightest shade, 100 to darkest shade) for each model, for the additional Scena

[Figure]

**Fig. 2.** Illustration similar to the figure for Setting 4 in the original main paper, but based on the adapted simulation setting suggested by Reviewer 1.

---

## Author Comment (AC2) · 2 May 2020

*We thank the three reviewers for their positive assessment and their thoughtful comments which, we believe, will strengthen the manuscript. Below we addressed each comment in turn. Our replies are in italics. We also created a track-changes PDF for your convenience.*

*Kind regards,*
*Sebastian Lerch, Sándor Baran, Annette Möller, Jürgen Groß, Roman Schefzik, Stephan Hemri and Maximiliane Graeter*

[Figure]

In this paper, a comprehensive simulation study is implemented, comparing multiple multivariate statistical post-processing methods which all combine standard univariate post-processing with one of four techniques to reintroduce spatial, temporal or inter-variable dependencies. It is well-written, an important contribution given the variety of techniques available and potentially very useful to identify optimal operational post-processing strategies for varying types of data. Just a few clarifications and changes are needed.

General comments:

I would have liked to have more focus (or at least comments) on the effect of ensemble size and dimension. The here chosen ensemble of 50 members is at the upper limit of what is now operationally produced, with the majority far below this number. Of course it is important for the study to have a sufficient number of data points so as to produce significant results, but it would also be interesting to look at settings with a smaller number of ensemble members. Also,the number of dimensions ranges from 4 to 5, which would correspond to looking at consistency between a few weather variables, but would usually be too low for a setting where preserving spatial or temporal features are important. I wonder if the findings would be different for smaller ensembles or higher dimensions.

*Thank you for this suggestion. Following a similar comment by Reviewer 1, we performed additional simulations to assess the effects of ensemble size. Please see the response to Reviewer 1 for a discussion of the role of the number of ensemble members.*

*To study the effects of the number of dimensions, we performed additional simulations for Setting 1 with dimensions between 2 and 50. Overall, the relative results are often not effected too much by changes in the number of dimensions, in particular in terms of the energy score. Somewhat more substantial differences can be observed*

*in terms of the variogram score. In a nutshell, GCA performs worse for higher dimensions, whereas ECC-S improves for larger values of $d$. The relative differences to SSh (in favor of SSh) become more substantial with increasing $d$, whereas the changes in relative performance of dECC show a strong dependency on the combined misspecification of variance and correlation of the raw ensemble.*

*We have added a paragraph to Section 4.2.1. Additional figures with results for $d = 2, 3, 10, 20, 30$ and 50 have been added to the Supplemental Material (Section 1.2 there). See the response to the corresponding comment by Reviewer 1 for a discussion of the display as multiple boxplots within one figure. Performing additional simulations to investigate different choices of $d$ for Setting 3 (which is Setting 2 in the revised manuscript) was impossible due to constraints regarding the computational requirements, see also our comment below.*

I find it very interesting that the performance of certain methods is sometimes very different when $\rho > \rho_0$ than in the opposite case. Do you have any explanation for this?

*We believe that despite the (relative) simplicity of the chosen simulation setup, interpreting differences in multivariate performance is challenging due to inter-connected contributions of misspecifications in mean, variance and correlation structure and their effects on the (still) not well understood multivariate proper scoring rules. We are thus unable to provide a comprehensive explanation for the particular role of the correlation parameters $\rho$ and $\rho_0$. Some potential explanations of differences in forecast performance may be given by observing that under certain circumstances, modifying a raw ensemble prediction by artificially weakening correlations, for example by randomly permuting ensemble member's forecast vectors, may unexpectedly improve predictive performance. This was for example observed in Schefzik (2017), and is likely an issue of underdispersed univariate forecast distributions that improve by the changes on the univariate forecast distributions imposed by the modifications of the correlation structure. However, to assess the effects of different sources of misspecifications in*

*further detail, additional multivariate verification techniques such as multivariate rank histograms should accompany the analysis. While such additional studies are beyond the scope of the paper, they represent an interesting starting point for future research.*

Specific comments

1. Line 71 and Line 153: The correlation matrix here is not necessarily the identity matrix, so I don't think it is a standard normal distribution.

*Fixed.*

2. Line 74: I would mention here that $m$ is the number of ensemble members

*Added.*

3. Section 2.2: There is a mixture of $x$ and $X$ used to define samples and ensemble forecasts, but I was confused why this distinction is made within the notation, it seems inconsistent.

*Throughout the description of the methods, we used $x$ to denote univariate quantities in the individual dimensions. $X$ in bold print is used to represent vector-valued quantities, and $X$ in normal print is used to for components thereof. A sentence has been added at the beginning of Section 2.2 to clarify this.*

4. Lines 184-186: For the other settings it is mentioned to which weather variables these settings could apply. It would be nice to add something like this to Setting 1, as well.

*A corresponding sentence has been added to the beginning of Section 3.1.*

5. Line 242: The notation in this setting is different from the others and this is confusing. Here, the forecasts and observations are marked with $x$ and $y$, whereas the other settings use $o/0$ to mark the observations.

*The notation has been changed accordingly, see also our answer to Reviewer 1.*

6. Line 276: Is there a specific reason, why $d$ is 4 in this setting and 5 in the others?

*The limitation to $d = 4$ was mainly due to computational requirements and numerical stability issues caused by the NORTARA package for $R$ used to generate the multivariate ensemble predictions and observations. A sentence has been added in the paper.*

7. Lines 292-293: Some of the matrices are in bold face, some are not.

*Fixed. Additional minor corrections to the description of Setting 4 (now Setting 3A) are detailed under "Further changes".*

8. Figure 1: I am a bit surprised to see that GCA is performing that much worse as compared to the other post-processing methods (in a univariate sense). There are even cases where the performance is equal or possibly worse than for the raw ensemble. Do you have an idea why that could be?

*In particular when compared to ECC-Q and related methods, we believe that this is mostly an issue of sampling. As discussed in Section 4.1, the univariate quantile forecasts of ECC-Q are (close to) optimal in terms of the CRPS, but the random samples from the predictive distributions obtained in GCA are not (see next comment). We have modified the corresponding sentence to make this more clear. Cases where performance is worse than the raw ensemble are very rare and likely arise when simulation parameters of the ensemble forecasts are close to these of the observations.*

9. Lines 328-330: Can you explain a bit further what you mean by "optimal in the terms of the CRPS"?

*When the CRPS is used for evaluating a forecast distribution represented by a finite simulated sample, the sample is implicitly interpreted as a set of quantiles from the underlying forecast distribution at levels $\frac{i-0.5}{m}, i = 1, \ldots, m$. Viewed differently, this implies that utilizing quantiles (at these levels) when generating a univariate sample from a forecast distribution will result in a lower expected CRPS than drawing samples at random. Even though the levels at which quantiles are obtained are not identical to the optimal quantile levels, the differences will be small for $m = 50$. Details on the mathematical background can be found in the referenced paper by Bröcker (2012). We hope that this becomes more clear with the modifications made to this part of the paper mentioned in the response to the preceding comment.*

10. Lines 334-335: Naturally, scenario D has the smallest improvement compared to the others. Does that also mean that the scenarios are on the same absolute skill level after post-processing?

*We did not further investigate univariate performance since we removed this paragraph following a suggestion of Reviewer 1.*

11. Footnote 4: In my opinion it would be clearer if you refer to ECC-Q as EMOS-Q in this section as well.

*We have modified the corresponding paragraph. Together with the new table comparing key features of multivariate post-processing methods (see comment by Reviewer 1), we hope that this sufficiently clarifies terminology.*

12. Lines 415-430: Can you refer to the figures in the appendix that show these results by number?

*References to the corresponding sections of the Supplemental Material have been
added.*

13. Line 446: "the VS might be better able to account.." This is confirming a known
result, therefore "might" is a bit unsuitable.

*Changed as suggested.*

Technical corrections

1. Line 327: I would move the sentence beginning "Note that" to footnote 4, as it directly
relates to the changes in the marginal distributions mentioned there.

*The corresponding paragraph has been modified following comments from Reviewer
1, and the footnote has been removed.*

2. Line 356: I find this sentence a bit confusing. Should there be a comma before "the
less information"?

*We have modified the sentence and added a comma as suggested.*

3. Line 386: Missing comma before "where"

*Fixed.*

4. Lines 415 and 441: I would add "parameter" after "observation location".

*Changed as suggested.*
* * *
2019-62, 2020.

---

## Author Comment (AC3) · 2 May 2020

This paper present results of an intercomparison study. Different multivariate ensemble post-processing methods are compared using toy-model simulations. The focus is on empirical copula methods that are generally applied as a second post-processing step,after univariate post-processing step, in order to provide coherent multivariate structure to ensemble calibrated forecasts.

As the reviewer is the developer of one of the methods compared here, he prefers to make himself know. There exists no conflict of interest (stricto sensu) but potential cognitive biases from the reviewer side when scrutinizing the results. The paper is

clear, well structured, and well written. However, the choice regarding the selection of illustrations is not sufficiently motivated to my opinion. This choice is important because it drives the discussion and the main conclusion of the study. A 3-point argument is developed below to explain this criticism.

You claim in the conclusion (L462/463) that the 4 simulation settings aim to mimic different situations and challenges occurring in practical situations. However, the link with practical situations is sometimes weak or missing. In particular, it would be interesting to link misspecification definitions with practical examples. What sigma $> 1$ and sigma $< 1$ mean, and similarly what does rho $>$ rho0 and rho $<$ rho0 mean and "look like" in practice? In which situations should one expect to encounter these types of misspecification? Which type of misspecification situations are the most common in practice? Are combinations of misspecified elements more common than others (sigma $< 0$ and rho $<$ rho0 for example)?

Once the link to the applications is clarified, illustrations could be chosen consistently. Result material is abundant, so one selection criterion could be to focus on the main misspecification encountered in practical situations. For example, you use sigma equals 1 to illustrate results in Setting 2. Does that often occur in practice? One could rather illustrate Setting 2 with sigma<1. Similarly, for Setting 3, scenario B is used for illustration purposes. It corresponds to the case where the ensemble forecasts have a heavier right tails and slightly higher point mass at zero than the observations. Does that often occur in practice? No justification for the use of this scenario as reference is provided. Scenario A (the ensemble comes from a distribution with smaller spread)or Scenario C (the observation distribution has a much heavier right tail) seem to be more likely to be faced in practice.

*Thank you for this helpful comment which sparked ample discussion between the authors about how to best diagnose and transfer misspecifications of real-world ensemble prediction system to the simulation settings.*

*From our experience with most standard surface weather variables, one would expect that the univariate ensemble predictions are typically underdispersive ($\sigma < 1$), and exhibit a bias ($\epsilon \neq 0$). Often, biases and dispersion errors will of course vary over time, and may be vastly different for different variables or geographical locations. The choice of the correlation parameters $\rho, \rho_0$ naturally hinges on the chosen correlation function for the simulation setting. In practice, this would likely correspond to an over-simplification of true correlations. In an attempt to diagnose realistic correlation parameters in the context of Setting 1, we have estimated correlation parameters for 2-day ahead ECMWF ensemble predictions of 00UTC temperatures at observation stations in Germany based on the 10-year dataset used in Rasp and Lerch (2018), as follows: For a randomly selected station, we choose the 13th, 26th, 38th, 50th closest station. If there are no substantial differences in altitude, we determine the empirical correlation among the observations at those 5 stations, as well as the correlation among each ensemble member's forecasts at those stations. Next, the differences between the estimated correlation in the ensemble members and the observation are computed for all 50 members. In addition, we determine the parameters $\rho, \rho_0$ from the exponential correlation function model assumed in the paper by numerical optimization. The procedure described above is repeated 100 times. Figure 1 shows differences in the empirical correlation, with histograms summarizing results over all 50 members and 100 repetitions. The differences in correlation are almost identical for all close stations, suggesting that the assumption of a fixed parameter $\rho$ and $\rho_0$ seems reasonable. Further, a similar conclusion can be obtained from Figure 2 which shows differences in the estimated correlation parameter of the exponential correlation model. Both figures suggest that correlations in the observations are over-estimated by the ensemble. Realistic settings thus probably relate best to simulation settings where $\rho > \rho_0$, but the values chosen in the paper (with differences of at least 0.15) appear to lead to possibly be too large differences (at least when compared to 2-day ahead temperature predictions over Germany).*

*In deciding on parameters for the simulation settings, we have mainly sought to cover a complete range of possible misspecifications rather than mirroring the situation in practice, in order to provide a more complete view of the performance of the multivariate post-processing methods. We have extended the discussion on the realism (or lack thereof) of the simulation settings in the Discussion, and have incorporated some of the arguments made above. Further, realistic values of the simulation parameters of Setting 1 that can be expected in practical applications are now discussed towards the end of Section 3.1. Setting 2 has been removed following the suggestion of Reviewer 1. Concerning the interpretation of the chosen scenarios in (former) Setting 3, for example Scenario B considers a situation where the forecast distribution estimates the amount of zero precipitation correctly, but otherwise the probability for obtaining a value smaller or equal to a fixed precipitation amount $x$ computed from the forecast distribution is always smaller than the corresponding probability computed from the distribution of the observation, see the right panel in Figure 3. In combination, these two features may not occur very often in practice, since one would expect that under-forecasting smaller precipitation amounts should come along with an underestimation of zero precipitation. See the left panel in Figure 3, showing fitted CDFs to a sample of observed precipitation values and corresponding forecasts of an individual ensemble member at a specific station in Germany based on real precipitation and ECMWF ensemble forecast data. Since a considerable number of scenarios with respect to the actual values of $\mu$, $\sigma$ and $\xi$ is conceivable in practice, depending on the climatological circumstances, further investigations based on real data are required to provide additional insights. Furthermore, the interplay between the 3 GEV0 parameters is specifically complex in the sense that all 3 parameters have a joint influence on the location and dispersion properties, so that simple misspecifications in mean and variance might correspond to various (different) sets of parameter combinations. However, we agree that a more detailed investigation of theoretical and practical properties of the GEV0 distribution is a highly interesting starting point for future research.*

Your conclusion points to the robustness of the SSh method and so you encourage post-processing practitioners to consider this method as their first choice. Based on the results presented in the manuscript and the ones in the supplemental material document, one could draw the opposite recommendation. First choice methods are available for different types of misspecification and so one could encourage to apply SSh only when the misspecification is unknown or difficult to identify. Specific recommendations would read:

1. If the ensemble is underdispersive and the ensemble correlation is too weak, d-ECC is the best option, both in terms of ES and VS.

2. If the ensemble is underdispersive and the ensemble correlation is too strong, ECC-S is the best option in terms of VS and one of the best options in terms of ES.

3. If the ensemble is overdispersive and the ensemble correlation is too strong, d-ECC is the best option, both in terms of ES and VS.

4. If the ensemble is overdispersive and the ensemble correlation is too weak, GCA is the best option in terms of VS, but is less performant in terms of ES. GCA proves to work well even when the overdispersiveness error characteristic is relaxed.

5. Otherwise consider using SSh, in particular when the correlation structures in theforecasts and observations are very dissimilar.

This is valid for all types of distributions (so all types of weather variables). Are these conclusions still valid in case of time varying misspecification? Verification results are missing to conclude here. Therefore, the authors are encouraged to investigate various sigmas in Setting 4 in order to collect evidences, and could draw conclusions accordingly

*Thank you for these suggestions. We agree that the conclusions may benefit from a more detailed discussion of situations where the different methods show advantages or disadvantages across settings.*

*With the changes and additional results in the paper and Supplemental Material following the comments by Reviewer 1 (Setting 2 was removed; Setting 4 (now Setting 3A) is accompanied by another variant, Setting 3B; and additional simulations with varying numbers of ensemble members and dimensions), we believe that it is difficult to arrive at general conclusions of this form that would be valid for all types of distributions and settings. In particular, the effects of misspecifications of the individual simulation parameters may be very different in, for example, the multivariate Gaussian distribution in Setting 1 and the censored GEV variant in Setting 3 (now Setting 2). In particular for the new time-varying setting, some of the results regarding the relative performance of dECC and ECC-S, respectively, differ somewhat from the recommendations summarized above. With Setting 1 in mind, realistic parameter choices to mimic properties of real-world dataset likely represent settings where $\sigma < 1$ and $\rho > \rho_0$. As you pointed out, these settings may favor ECC-S over ECC-Q.*

*We have modified the discussion in Section 5 to provide more detailed suggestions and conclusions regarding the overall performance of the methods.*

[Figure]

**Fig. 1.** Differences in empirical correlation of observations and ensemble members at a set of 5 close stations based on the dataset of Rasp and Lerch (2018).

[Figure]

**Fig. 2.** Differences in estimated correlation of observations and ensemble members according to the exponential correlation function model assumed in the simulation settings at a set of 5 close stations based

[Figure]

**Fig. 3.** Cumulative distribution functions of $\text{GEV}_{0}$. The fit parameters in the left panel are $\mu_{0} = -1.0698, \xi_{0} = 0.2700, \sigma_{0} = 1.9906$ for the observation (red line) and $\mu = 0$